# Planning for sustainable cities by estimating building occupancy with mobile phones

Edward Barbour[1,2,3], Carlos Cerezo Davila[4], Siddharth Gupta[1], Christoph Reinhart[4], Jasleen Kaur[5] & Marta C. González [1,3,6]

Accurate occupancy is crucial for planning for sustainable buildings. Using massive, passively-collected mobile phone data, we introduce a novel framework to estimate building occupancy at unprecedented scale. We show that, at urban-scale, occupancy differs widely from current estimates based on building types. For commercial buildings, we find typical occupancy rates are 5 times lower than current assumptions imply, while for residential buildings occupancy rates vary widely by neighborhood. Our mobile phone based occupancy estimates are integrated with a state-of-the-art urban building energy model to understand their impact on energy use predictions. Depending on the assumed relationship between occupancy and internal building loads, we find energy consumption which differs by +1% to −15% for residential buildings and by −4% to −21% for commercial buildings, compared to standard methods. This highlights a need for new occupancy-to-load models which can be applied at urban-scale to the diverse set of city building types.

[1] Department of Civil and Environmental Engineering, MIT, Cambridge, MA, USA. [2] Centre for Renewable Energy Systems Technology, Loughborough University, Loughborough, LE, UK. [3] Lawrence Berkeley National Laboratory, Berkeley, CA, USA. [4] Sustainable Design Lab, MIT, Cambridge, MA, USA. [5] Signify Research North America (formerly Philips Lighting), Cambridge, MA, USA. [6] Department of City and Regional Planning, UC, Berkeley, CA, USA. Correspondence and requests for materials should be addressed to M.C.G. (email: martag@berkeley.edu)

Energy use in buildings accounts for over 40% of total primary energy use in the U.S and E.U.[1,2], and is particularly concentrated in urban areas, which are set to rapidly expand in the near future[3]. Efficiency measures applied to urban buildings therefore constitute an unparalleled opportunity for global energy use and emissions reduction[4,5]. Recognizing this, many national and local governments have implemented or are considering policies to promote building efficiency, for example, by incentivizing increased insulation levels or high efficiency appliances. However, these initiatives typically operate with limited resources, both in terms of budget and technical assistance. Urban-scale building energy models have therefore been developed to support strategic decision-making by pinpointing policies with the greatest saving potential.

Urban-scale energy models can be broadly split into two categories; those that rely on data-driven statistical or machine learning models (black-box models) and those that include a physics-based building simulation engine (white-box models). In the first category, statistical models typically reduce the energy consumption of a large number of buildings to a small number of explanatory variables[6–8]. These models can subsequently be used for energy benchmarking and were the first to estimate urban-scale impacts of potential Energy Conservation Measures (ECMs). However, while useful, these models have two notable limitations. Firstly, the lack of detail regarding individual buildings can result in poor out-of-sample predictions, particularly related to the widespread impacts of many ECMs and feedback between ECMs and building operation[8], and secondly, they cannot be used in the design of new, or densification of existing, urban districts required to facilitate booming urban population growth[9].

In the second category, a new class of Urban Building Energy Models (UBEMs) has recently been developed, which includes detailed physics-based simulations of thousands of individual buildings in cities or districts[10–16]. In an UBEM, each individual building is represented by a dedicated physics-based engineering model, the likes of which are commonly used worldwide for engineering design, code-compliance demonstration, and improved operation[9]. UBEMs rely on highly automated workflows to generate the detailed individual building models without requiring time-intensive work by building modeling experts[9,17,18], often combining several datasets, including Geographic Information Systems (GIS) databases, LiDAR and stock building archetypes. The models are then calibrated to match real data samples[11,19]. However, despite high versatility[20], inaccuracies are introduced in these models due to a lack of understanding regarding building occupancy at an urban scale[20–22].

For most buildings, with the exception of certain industrial facilities, occupant presence and behavior have a decisive impact on building energy use. In recognition of this well-known fact, the International Energy Agency (IEA) has funded two Annexes dedicated to understanding occupant presence and behavior in buildings and improving models, Annex 66 and its follow-up, Annex 79[23,24]. While at the individual building level, several occupancy models exist (i.e., refs. [25,26].), only a few studies have considered occupancy in an urban context[27]. In absence of better data, standardized deterministic space-based occupant presence and behavior models are the only viable approach for UBEMs of urban areas with mixed-use buildings[20].

To understand building occupancy at an urban-scale requires either a vast array of sensing infrastructure or knowledge of population movements over an entire metropolitan regions–capturing the daily movements of citizens between different buildings. To that end, in this work, we propose using massive, passively-collected mobile phone data to infer building occupancy on a city-level. This data has already been used to extract locations where individuals stay (stay points), as well as to infer their location-based activities[28] and for highly-aggregated electricity load predictions[29,30]. A recent modeling framework (TimeGeo) synthesizes previous findings in human mobility[31,32], demonstrating that sparse mobile phone data can be used to model individual trajectories for entire urban populations[33]. Several other passive data sources have already been used to understand city dynamics[34] and infer building occupancy[35–39], in particular bluetooth, wifi, cameras and electricity data, as well as their combinations. However, in contrast with mobile phone data, these sources are not available at sufficient scale for predicting simultaneous occupancy for thousands of different buildings.

In this work, we develop a method for estimating building occupancy at urban-scale by extending the TimeGeo framework[33]. Our statistical method assigns occupants to buildings and we demonstrate the proposed framework for 83,000 buildings in the city of Boston, using the individual trajectories of 3.5 million urban inhabitants. The building assignment is probabilistic and analogous to route assignment models, successfully developed with mobile phone data in vehicular traffic[40,41]. In order to demonstrate the significance of our results for energy modelling, we develop an UBEM of a central, mixed-use neighborhood and compare occupancy estimates between standard reference methods and the mobile-inferred building occupancy. Since occupancy is a principal driver of building energy use but no model relating occupancy and building loads on this scale exists, we develop approximate upper and lower bound scenarios for the impact of occupancy. Firstly, a low-impact of occupancy scenario wherein we adjust the occupant-driven reference building loads according to the relative occupancy, primarily changing the timing of loads rather than the magnitude. Secondly, a high-impact scenario, wherein we adjust the occupant-driven loads according to the absolute mobile-inferred occupancy, significantly changing both load magnitude and timing. We find that these scenarios imply energy consumption that differs by +1 to −15% in residential buildings and by −4 to −21% in commercial buildings. Finally, we illustrate that the mobile occupancy implies the impact of energy efficiency measures may be mispredicted, finding savings from insulation improvements up to 25% higher than predicted with standard occupancy while savings from improving equipment efficiency were predicted up to 40% lower in our modelled region. Our method therefore provides an improved alternative estimate of the number of occupants in buildings, in contrast to indirect space-based estimates which represent standard practice in city-scale energy planning today.

## Results

**Reference building occupancy.** In the United States, the Department of Energy (DOE) reference commercial buildings dataset[42] provides the standard set of templates for the US building stock for the EnergyPlus simulation engine[43,44]. These include sixteen different building categories in all US climate zones with different construction periods. Each building category has its own associated fractional occupancy and load schedules based on previous studies by the DOE, ASHRAE and several US National Research Labs. The schedules include estimates for occupant density based on building type (i.e., small retail occupancy: 0.141 occupants per $m^2$ of floor area) and peak load values per unit floor area for different load types (i.e., office equipment density: $17 \, Wm^{-2}$)[45]. When used for individual buildings, the space-based templates can be adjusted using local knowledge of the occupants; however, this is not possible at an urban-scale[20]. Hence, without any empirically based alternatives, city-scale energy models typically use the reference building occupancy, introducing uncertainty into energy-use predictions and, as a result, into predicted savings from efficiency measures[20]. This barrier for urban energy policy development is an international

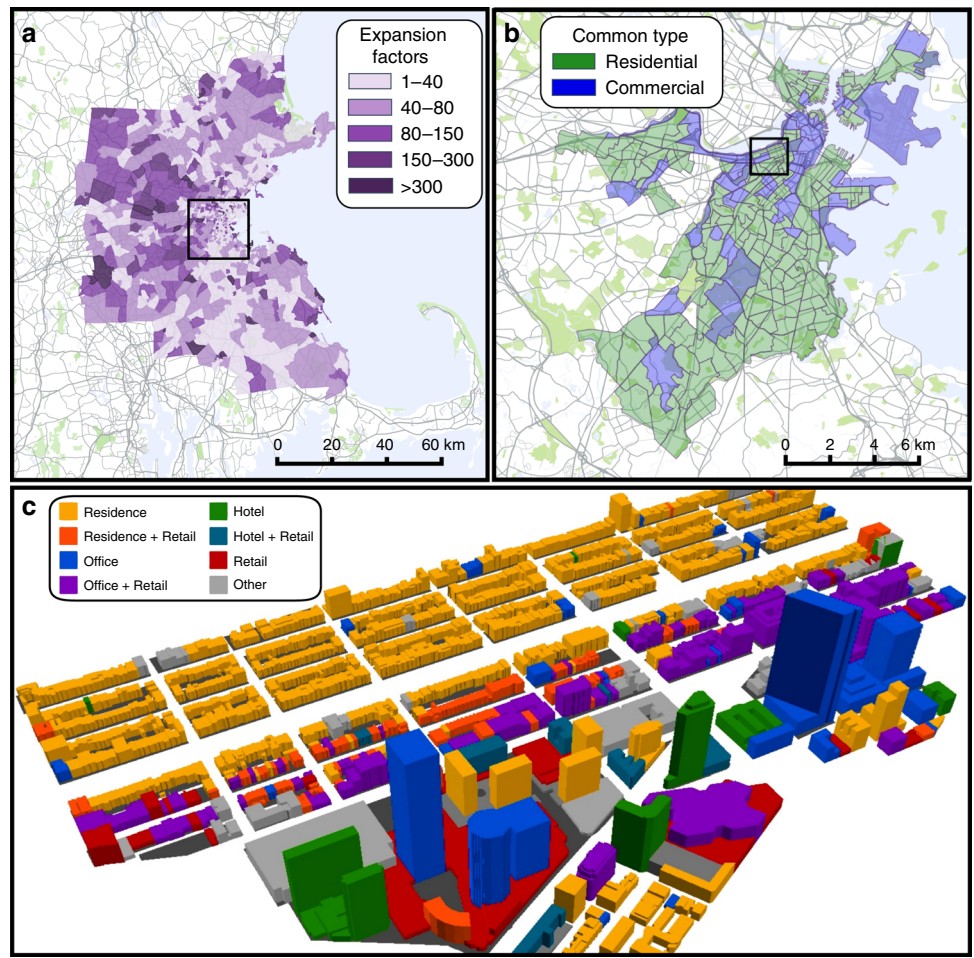

**Fig. 1** Extent of the data. **a** The extent of the TimeGeo simulation in the greater Boston metropolitan area. Each census tract is colored by its expansion factor. **b** The extent of the Boston Buildings dataset—tracts are marked as either majority residential or majority commercial depending on the most common building use. **c** The buildings used in the Urban Building Energy Model, colored by type. These buildings occupy five census tracts. Black boxes in Fig. 1a. and Fig. 1b. illustrate the extent of the next panel. Maps created using Google Maps, 2019 Google

issue, since all national building stock models suffer from the same lack of empirical occupancy data.

**From mobile phone stay points to building occupancy.** Using the TimeGeo framework[33] we simulate the individual mobility of Boston's entire metropolitan area population at high resolution. The framework starts with the Call Detail Records (CDRs) of 1.92 million anonymous mobile phone users of for the period 20 February till 30 March 2010 in the Greater Boston area (see Methods). Stay points for these users are identified based on consecutive mobile phone records within certain temporal and spatial thresholds, namely 10 min and 300 m. Each stay point is characterized as either home, work or other, depending on whether it is estimated to occur at the individual's place of residence, work or some other location[33], and includes a start time, duration, latitude/longitude coordinate pair and user id. We find that nearly 200,000 users have more than 50 total stays and at least 10 home stays (home stays only occur at an individual's specific home-designated building) during the observation period. These are designated active users and their phone records are used to extract the mobility parameters for an empirically based population-wide mobility model, reliant only on measurable parameters in the data. In each census tract in Boston's metro area, TimeGeo expands the active phone users to the population (i.e., simulating 3.54 million people including 2.10 million workers and 1.44 million non-workers). The model encompasses

individuals with homes in the region shown in Fig. 1a. The colors in Fig. 1a indicate the average expansion factor of commuters and non-commuters for each tract, calculated as the ratio of the total tract population (as defined by the 2010 census) to the number of active mobile users with homes within that tract. The model has been proven to be accurate at the census tract level[33] in comparison with both the 2009 National Household Travel Survey[46] and the 2010–2011 Massachusetts Travel Survey[47] (see Supplementary Fig. 1 and Supplementary Note 1). TimeGeo offers significant improvements on current urban mobility models, which typically involve expensive surveys and have very low sampling rates. It should be noted that the model is primarily limited by the extent, resolution and duration of the data. As such, as more large-scale data with higher frequency (i.e., GPS traces) and longer observation periods (i.e., years) become available, the model resolution could be increased and heterogeneity improved[33].

The fine-scale population mobility model has been shown to agree with existing methods of quantifying population mobility at the census tract level[33]. Hence, each stay can be thought of as a person-visit to a tract, as implied by the empirical mobile phone data and the Boston metro census population, and in agreement with current best-in-class travel demand models. Figure 2a illustrates the journeys for the people who visit the five tracts shown in Fig. 1c. The predicted tract-level occupancy is revealing, along with the expected flux of people in-and-out-of the area, as

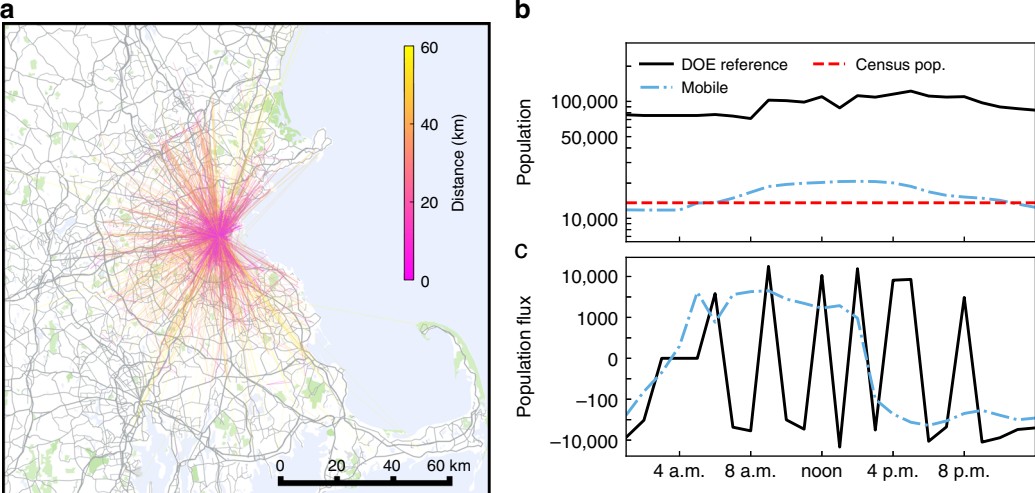

**Fig. 2** Residents and visitors in the Back Bay. **a** Predicted journeys from the TimeGeo for users who visit the five tracts shown in Fig. 1c. Each line represents approximately 50 users. **b** The total predicted hourly occupancy for the area shown in Fig. 1c according to the TimeGeo model and as implied by the DOE reference building model. The census population for the area is also shown. **c** The predicted population flux as implied by the reference buildings and TimeGeo model. Map created using Google Maps, 2019 Google

shown in Fig. 2b, c. Here, we see that the implied occupancy of the five tracts over the course of a typical weekday using DOE reference building models ranges from 71,300–122,500, whereas the mobile-informed model ranges from 11,800–20,700. Given the census population for these five tracts is 13,600, the mobile model certainly seems more credible and the DOE reference occupancy standard may be overestimating actual occupancy numbers by over 600%.

Each tract typically encompasses hundreds to thousands of buildings, for example, the five tracts used in the energy model (Fig. 1c) contain 1330 buildings. Therefore, to assign each stay to a building we assume that each stay point represents a visit to a building in the same tract and probabilistically map each stay to a building. Buildings data is provided by the city of Boston and includes footprint, geometry, height and a tax assessment type (see Supplementary Fig. 2 and Supplementary Note 2). We re-classify the buildings into three broad classes; residential, commercial and industrial (see Methods) and stipulate that home type stays can only be assigned to residences, while work stays can be assigned to all building types except residential and other stays can be assigned to commercial buildings only. Therefore, we assume that an individual stay point can potentially be located in any building within the same tract, provided the building is open for the stay duration and the building functionality matches the stay type. To probabilistically select a particular building, we use nominal building capacities, and assume that, in general, buildings with higher capacity attract more stays. To estimate the nominal capacities we use typical Per Capita Area (PCA) values for each functional building type (see Methods, Supplementary Table 1 and Supplementary Note 3). We assume residential buildings are always open and have a PCA of 40 m$^2$, i.e. a house with a floor area of 200 m$^2$ would have a nominal capacity of five occupants. For non-residential buildings, we use Places Of Interest (POIs) available in digital maps where the exact building functional use is unknown (Fig. 3b, see Methods, Supplementary Table 2 and Supplementary Note 4).

Furthermore, we hypothesize that at a city-block level there may be a rich-get-richer effect, which results in popular areas attracting more people per-unit-floor-area. Therefore, in our model, stays are preferentially attracted to areas with larger capacities for that type of stay. This is based on the observation

that shops and industries agglomerate due to clustering of economic activities[48]—for example, resulting in the formation of popular shopping, dining or entertainment districts. To model this, we adopt a two stage process when assigning a stay to a building within a particular tract. Firstly, we spatially group the possible buildings by their centroid coordinates using hierarchical agglomerative clustering and Wards minimal increase of variance method[49] to form city-blocks. Then, after probabilistically selecting a block we select an individual building (Fig. 3c). The preferential attraction at the cluster level is given as follows:

$$P(i) = \frac{\left[\frac{C_i}{\sum_{i=1}^{N} C_i}\right]^{1+\mu}}{\sum_{i=1}^{N}\left[\frac{C_i}{\sum_{i=1}^{N} C_i}\right]^{1+\mu}} \quad (1)$$

For a given cluster $i$ with a nominal capacity $C_i$, $P(i)$ is the probability of that cluster being assigned a particular stay. The total capacity of each cluster $i$ is the sum of the capacity of the $M_i$ buildings contained within-cluster $i$, and the total tract capacity is the sum of the capacity of the $N$ clusters. Therefore, $C_i = \sum_{j=1}^{M_i} C_{ij}$, where the buildings are indexed by $j$, and $C_{ij} = \alpha_{ij} A_{ij}$. $\alpha_{ij}$ and $A_{ij}$ are the PCA and total floor area of building $j$ in cluster $i$, respectively. The parameter $\mu$ varies the degree of preferential attraction to areas with a high-capacity for stays of that type (see next section). Once a cluster $i$ has been selected, then within that cluster we assign the stay to a building $j$ with probability $P(j|i)$, proportional to its relative within-cluster capacity, as described by Eq. (2).

$$P(j|i) = \frac{C_{ij}}{\sum_{j=1}^{M_i} C_{ij}} = \frac{C_{ij}}{C_i} \quad (2)$$

Figure 3 illustrates the process of assigning stays to the buildings. Once we have considered all the stay points within a particular tract for the 24 h period, we assume that we have a complete picture of building occupancy over the day.

**Occupancy for Boston's buildings**. We now study the expected building occupancy for $0 \leq \mu \leq 1$. The rationale for adopting these bounds are, (1) we do not expect a rich-get-richer effect where the

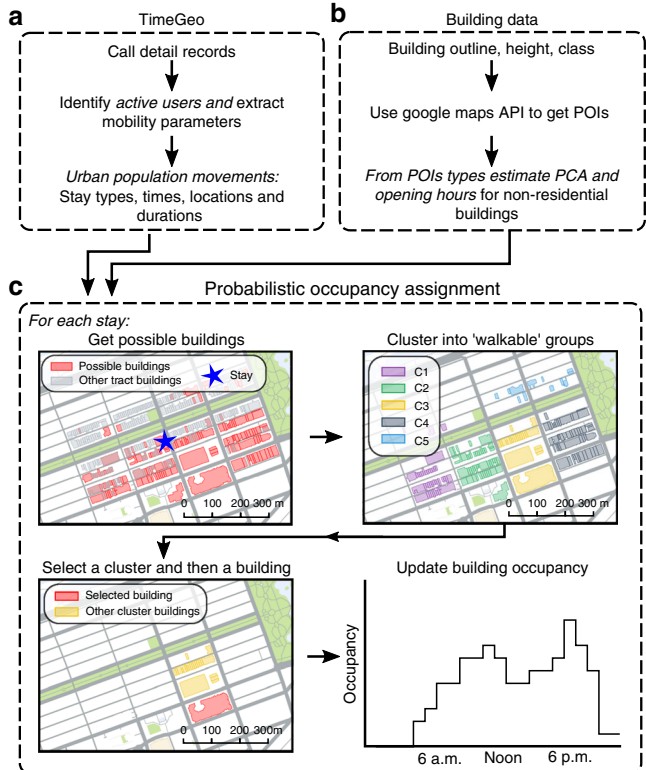

**Fig. 3** CDR data and building data to building occupancy method. **a** The TimeGeo framework going from call detail records to population-wide stay locations[33]. **b** The building data includes building type, PCA and opening hours. **c** The set of possible buildings for a stay depends on building type and opening hours (for the stay type other as shown, possible buildings are open commercial buildings). Buildings are clustered into localized spatial groups using Ward's minimum increase of variance method and the dendrogram is truncated at a Ward distance of 500 m, which yields city-block shaped clusters. Then, a cluster is probabilistically selected followed by a building, and the stay is added to that building's occupancy profile. Maps created using Google Maps, 2019 Google

returns are diminishing (i.e., if $\mu < 0$ then areas with more opportunities (capacity) will attract proportionally less people per unit floor space) and (2) we do not anticipate that $\mu > 1$ because above this value we find that high-capacity areas start to dominate the stay attraction to an unreasonable degree, leaving many unoccupied buildings. The effect of the non-linear parameter $\mu$ is discussed in detail in Supplementary Note 5 (see also Supplementary Fig. 3 and Supplementary Table 3).

The results are consistent with residences having high night occupancy and commercial buildings having higher daytime occupancy, as people generally leave their homes during the day (Fig. 4a). Figure 4b shows that greater $\mu$ results in more highly occupied spaces, shifting occupants from lower occupancy spaces to higher ones. In the range $0 \leq \mu \leq 1$ we see that shift in the distributions change is slight and the median occupancy-over-capacity ratio for residential buildings decreases from 1.4 to 1.2 while the median commercial occupancy decreases from 0.3 to 0.2. For the 83,000 buildings in the city of Boston, we find that most commercial buildings have nominal capacities that are significantly higher than their peak occupancy (Fig. 4b shows that mobile-inferred peak occupancy is higher than the capacity in only 5% of commercial buildings). Conversely, for residential buildings, the occupancy-over-capacity distribution has a long tail and implies a wide range of residential occupant densities. These are highly neighborhood dependent and we observe high

numbers of occupants in student areas situated in close proximity to universities, where higher-than-average residential occupancy persists throughout the day (Fig. 4a and Supplementary Fig. 4). For commercial buildings, our estimate of peak capacity is rarely approached. We find that when $\mu = 0.5$, the most common value for residential peak-occupancy-over-capacity is 1 while the median is 1.3. Therefore our assumed residential PCA appears close to typical values for residential space in Boston.

**Occupancy for urban energy predictions**. At this point, we create an urban energy model for the mixed-use district in the Back-Bay region of Boston, covering five census tracts with 1,330 buildings as shown in Fig. 1c. Out of these, we were able to model 1,266 buildings using the DOE reference buildings (see Supplementary Fig. 5). The other buildings were considered in the occupancy-assignment process (since they could still be open and attract people); however, due to atypical uses (i.e,. fire/police, churches, etc.), no appropriate reference model existed and therefore these were not included in the energy model. To generate the modeled buildings we used data regarding use per floor, period of construction and geometry. We use the previously developed UMI framework[10] (see Methods) which is based on EnergyPlus, a state-of-the-art whole building energy simulation engine developed by the US DOE[43]. The building constructions and systems were specified through archetype templates based on the DOE Reference Buildings dataset (see Supplementary Note 6 and Supplementary Figs. 6 and 7). The templates have the same nominal peak occupancy values as used for the 83,000 building analysis and also contain 24-h occupancy profiles dependent on the building use. The occupancy is compared to our mobile-informed occupancy results, as shown in Fig. 4d. Our results suggest that in individual commercial buildings the reference building occupancy can over-predict occupancy by an order of magnitude or more at all times during the day, in our modeled region.

The reason for the major discrepancy between the reference building occupancy and mobile-inferred occupancy is that the mobile occupancy is normalized to the Boston metro area population, whereas the reference building occupancy profiles are based on a combination of factors including ASHRAE ventilation requirements, fire safety codes, and generalized building-manager-surveys. Importantly, these factors have no ability to account for the city-specific population and its movements, and therefore, as shown in Fig. 2b these are inconsistent with the census population. Accordingly, for the total modeled neighborhood we find the mobile occupancy corresponds to only 15–24% of the reference building occupancy over the course of a typical weekday. The population fluxes also seem unreasonable whereas the mobile-inferred people-fluxes between different regions are consistent with current transportation models[33].

**Occupant-driven energy loads**. Since occupancy is often a large driver for building energy use[8,20,37,50], the differences between the reference building and the mobile-inferred occupancy imply different energy consumption patterns. Unfortunately, while methodologies exist that can predict appliance use for individual buildings based on the maximum number of occupants[26], no satisfactory method exists that can function at urban-scale[20,50,51]. Therefore, we rely on the occupant-driven loads prescribed in the DOE reference buildings, which are empirically related to the reference occupancy. We develop two rule-based scenarios, which broadly represent high-impact-of-occupancy and low-impact-of-occupancy case studies, and compare the results of each with a base model using the DOE reference buildings. The resulting three scenarios are described as follows. First, the DOE reference

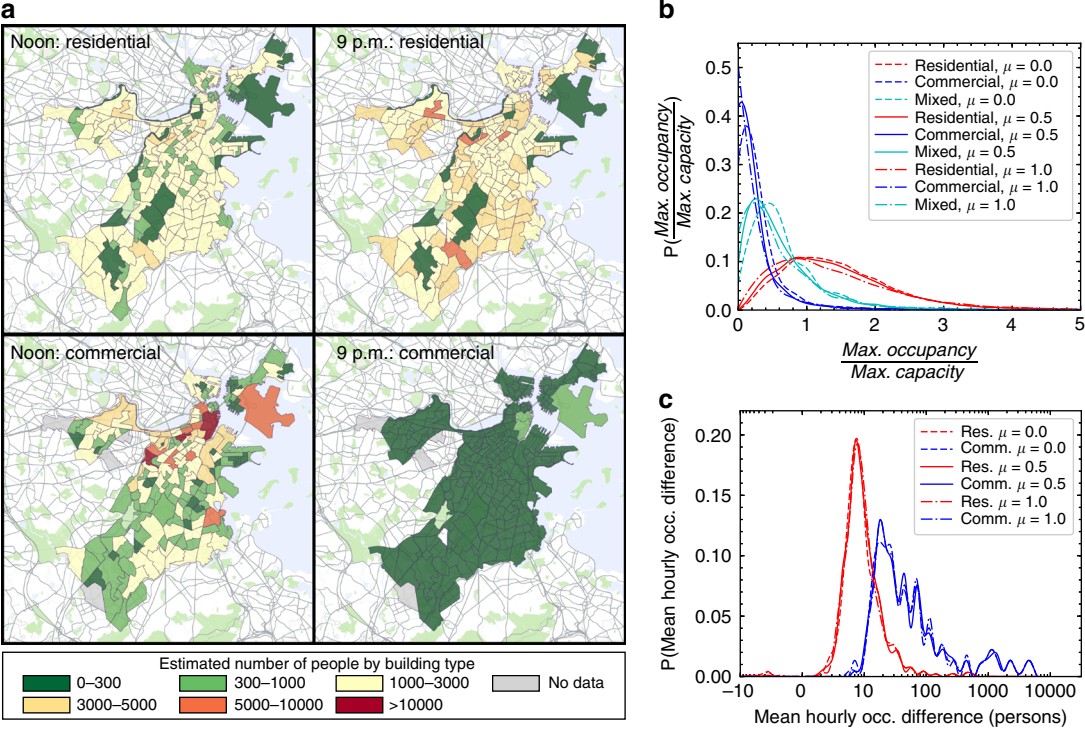

**Fig. 4** Building Occupancy results. **a** Estimated number of building occupants by building type for the mobile-inferred model for residential and commercial buildings. **b** Distributions of the ratio of maximum observed occupancy to nominal building capacity for ~83,000 buildings in Boston. **c** Distribution of average hourly occupancy difference between DOE reference and mobile-inferred occupancy for the 1,266 buildings in the energy model. Positive means the DOE occupancy is greater while negative implies the opposite. Maps created using Google Maps, 2019 Google

scenario is the control scenario and presents simulations using occupancy and load schedules unaltered from DOE reference building templates. Second, the low-impact scenario develops occupancy-driven load schedules considering the ratio between the relative mobile occupancy and relative reference building occupancy, as described in Eq. (3). Third, the high-impact scenario develops occupancy-driven load schedules considering the ratio between the absolute mobile occupancy and reference building occupancy, as described in Eq. (4).

$$L^{\mathrm{mob}}(h) = b + (L^{\mathrm{DRB}}(h) - b)\frac{\frac{O^{\mathrm{mob}}(h)}{\mathrm{MAX}(O^{\mathrm{mob}})}}{\frac{O^{\mathrm{DRB}}(h)}{\mathrm{MAX}(O^{\mathrm{DRB}})}} \quad (3)$$

$$L^{\mathrm{mob}}(h) = b + (L^{\mathrm{DRB}}(h) - b)\frac{O^{\mathrm{mob}}(h)}{O^{\mathrm{DRB}}(h)} \quad (4)$$

In Eqs (3) and (4), $L^{\mathrm{mob}}(h)$ is the fractional schedule for each load at hour $h$ associated with the mobile occupancy, $O^{\mathrm{mob}}(h)$. $L^{\mathrm{DRB}}(h)$ is the DOE Reference Building load schedule associated with the reference building occupancy profile $O^{\mathrm{DRB}}(h)$ and $b$ is the baseload—the proportion of load independent of occupancy —which we assume is given by the minimum value of $L^{\mathrm{DRB}}(h)$.

The purpose of the scenarios is to create relationships between occupancy and the resulting use of lights, equipment and hot water. In both, we define a constant baseload which includes non-occupant related energy use, such as refrigerators etc. For the remaining loads we assume upper and lower bounds as follows: For the lower bound (low-impact), we assume that loads vary by the relative ratios between mobile and reference building occupancy, so if $\frac{O^{\mathrm{mob}}(h)}{\mathrm{MAX}(O^{\mathrm{mob}})} > \frac{O^{\mathrm{DRB}}(h)}{\mathrm{MAX}(O^{\mathrm{DRB}})}$ the predicted load increases. This primarily corrects time-of-day effects, i.e., if occupants arrive earlier loads are shifted to be earlier. In contrast, the upper bound (high-impact) assumes non-baseload loads change as the absolute

ratio between the mobile and reference building occupancies, i.e., if the mobile occupancy predicts half the number of people at a certain time the (non-baseload) loads will be halved. The high-impact scenario works best in a compartmentalized building, such as an apartment block, where light and equipment scale with the number of occupied rooms at any moment in time. Conversely, the low-impact scenario is suitable for open plan offices or retail, where occupants drive when lighting, fans etc. are switched on, but the exact consumption is insensitive to the absolute occupancy. However, Eq. (3) leads to implausible results if the occupant number predicted by the mobile occupancy schedule at a particular time is much lower than the reference building schedule and simultaneously corresponds to a larger relative mobile occupancy. Therefore, we stipulate that the predicted mobile load can only be a factor of $\frac{O^{\mathrm{mob}}(h)}{O^{\mathrm{DRB}}(h)}$ higher than the reference building load. Supplementary Fig. 8 illustrates the occupancy-driven load schedules for different types of building under each scenario (see also Supplementary Note 7).

Heating and cooling may also be occupant-driven—occupants may change temperature set-points according to personal thermal comfort and reduce heating/cooling in unused space. EnergyPlus specifies heating and cooling set-points and requires a schedule for ON/OFF operation. In all scenarios, we use the consistent set of heating and cooling set-points provided in the DOE reference buildings dataset. The differences in heating and cooling loads between the scenarios is then driven by the number of occupants and their appliance usages.

**Urban energy prediction with mobile occupancy.** Using the previously described UBEM we now run simulations for the 1,266 modeled buildings under each of the three scenarios (DOE reference, low-impact and high-impact), each with occupancy distributions for $\mu = 0$, $\mu = 0.5$, and $\mu = 1$. We simulate one

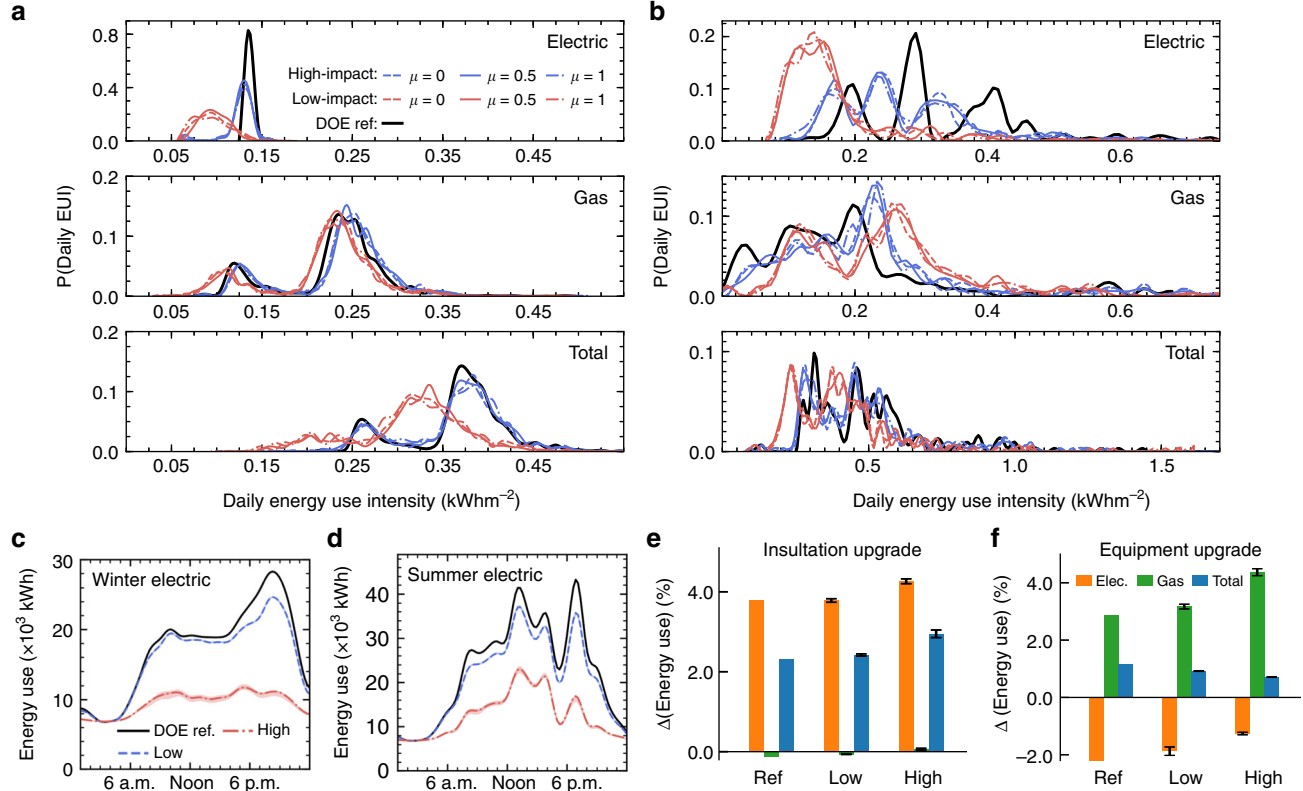

**Fig. 5** Scenario energy consumption predictions. **a** Distribution of the daily EUI for all residential-only buildings for $\mu = 0$, $\mu = 0.5$, and $\mu = 1$ occupancy. **b** Distribution of the daily EUI for all non-residential-only buildings. **c** Hourly electricity use results for winter in each scenario (shading denotes the range from the different $\mu$ values). **d** Hourly electricity use results for summer in each scenario. **e** Median change in energy use (original minus upgrade) from upgrading the insulation effectiveness in residential spaces under each scenario. Error bars illustrate the range of predictions for different $\mu$ values. **f** Median change in energy use from upgrading the equipment efficiency in commercial spaces under each scenario

representative day in each season using the Typical Meteorological Year (see Methods) to get a predicted daily energy consumption value for each building in each season. For the reference model we also compare the predicted annual Energy Use Intensity (EUI)—the energy consumption per unit floor area —for each building to the national average value for the equivalent building types in the Commercial Building Energy Consumption Survey and the Residential Building Energy Consumption Survey. Here, we find the average difference of the modeled EUI compared to the CBECS/RBECS averages for the specific geographic region ranged between 5% and 20% for the modeled building types with the simulated value always being higher than the CBECS ref.[19].

Figure 5a, b shows the predicted distributions of daily Energy Use Intensity (EUI) averaged over all seasons, grouped into residential-only and commercial buildings, for each $\mu$ value. Consumption is further split into gas (heating and hot water) and electricity (cooling, equipment, and lighting) estimates. We find that the low-impact scenario predicts increased gas use and decreased electricity use compared to the DOE reference scenario, with both effects being larger in commercial buildings. The high-impact scenario exaggerates both of these effects in commercial buildings, although for residential buildings gas use is decreased. The uncertainty from the different occupancy distributions (as generated by the range of $\mu$ values) is also illustrated. We see that the differences between the occupancy distributions are insignificant compared to the difference between the high-impact and low-impact scenarios.

In comparison to the DOE reference scenario, the low-impact scenario predicts a median EUI increase of $0.6 \pm 0.1\%$ for residential buildings and a decrease of $4.2 \pm 0.1\%$ for commercial buildings. The high-impact scenario predicts median EUI decreases of $14.9 \pm 0.9\%$ and $21.0 \pm 0.4\%$ for residential buildings and commercial buildings, respectively. Figure 5c, d illustrates hourly electricity consumption predictions for all scenarios aggregated for all modeled buildings. Electrical load profiles and total winter gas usage are particularly important to grid operators and utilities to plan peak electrical generation capacity and gas storage respectively. We see that the prominence of the winter electric peak is reduced by 4 MW in the low-impact scenario and almost completely removed in the high-impact scenario. The summer peak electricity loads are also predicted significantly lower, due both to reduced numbers of occupants and heterogeneous occupancy. These effects are overemphasized in the high-impact scenario. Total winter gas use is predicted ~9.9% greater in the low-impact scenario while it is 10.4% greater in the high-impact scenario (see Supplementary Tables 4 and 5, Supplementary Note 8 and Supplementary Figs. 9 and 10).

**The impact of energy efficiency measures.** Finally, we predict the impact of two different generic energy efficiency interventions, that could be encouraged by energy policy (see Supplementary Note 9). Firstly, we improve wall and roof insulation by 10% in comparison to the 2010 building code requirements for all residential spaces, and secondly, we implement a 10% efficiency gain for equipment in commercial spaces (plug loads not including light and heating or cooling). Figure 5e, f shows the

median reduction for each efficiency measure in each scenario. The DOE reference scenario predicts upgrading insulation decreases the median gas use by around 3.8% but increases electricity by 0.1%, which results in a median decrease in total energy use for residential buildings of ~2.3%. While the effects are similar for the low-impact scenario, the high-impact gas saving increases to 4.2%. Interestingly, electricity consumption decreases by 0.1% in the high-impact scenario, having increased by 0.1% in the DOE reference and low-impact scenarios. With upgraded equipment efficiency, Fig. 5f shows that electricity use is decreased while gas use increases to compensate for the reduction in the heat generated by equipment. The DOE reference scenario predicts the median commercial building saves approximately 2.9% in electricity consumption, which is improved to 3.2% in the low-impact scenario and to 4.4% in the high-impact scenario. However, due to increased gas use, the median total energy savings for commercial buildings are 1.2, 0.9, and 0.7% in the DOE reference, low-impact and high-impact scenarios, respectively (see Supplementary Tables 6 and 7). These results show that knowledge of occupancy is crucial to accurately predict the results of large-scale efficiency improvements in building envelopes or systems.

## Discussion

In this paper, we have presented a framework for improving urban scale building occupancy estimates. Our method extends proven techniques for modeling urban mobility based on mobile phone and census data and is the first to estimate urban-scale building occupancy with mobile phone data. We find that typical maximum daily occupancy in individual commercial buildings is likely to be only 20–30% of assumed capacity by building type. Conversely, residential occupancy is highly neighborhood dependent, with certain areas experiencing much higher occupancy per unit of floor space than others. The differences between our mobile phone based occupancy estimates and current occupancy assumptions based on building types arise because current methods treat buildings in isolation whereas our estimates consider that occupants can visit multiple buildings, i.e., people leave residential buildings to go to office buildings to work. Therefore our estimates are normalized against the total urban population. While the exact results are specific to the modeled region, the method is easily portable to other locations due to the ubiquitous nature of the data sources used.

These differences between the estimated mobile occupancy and current building-based assumptions have strong implications for urban energy modelling and predictions. In a state-of-the-art UBEM of a mixed-use urban neighborhood in Boston, the mobile-estimated occupancy predicted energy consumption that differed by +1 to −15% for residential buildings and was reduced by 4–21% for commercial buildings compared with reference building model. This is highly consequential for urban energy policy development, since policy makers rely on urban energy models to inform their decisions and these decisions have widespread and long-term impacts. For example, in the US, entire new neighborhoods are designed with the US Green Building Council's LEED (Leadership in Energy and Environmental Design) certification, which acts as a de facto building standard in certain jurisdictions and for which compliance is demonstrated through a simulated comparison against a code compliant base version[45]. Furthermore, urban energy models will be required to develop future sustainable solutions specific to individual cities or neighborhoods[52].

Given the large range between the low-impact and high-impact of occupancy scenarios, our work underscores an urgent need to understand how occupancy drives different load classes[53,54] across the diverse set of building types found in cities. Our results

suggest that building heating and electrical loads may be systematically mispredicted due to incorrect occupancy assumptions, especially in large commercial buildings. However, while our results imply heating loads may be under-predicted in most buildings due to lower-than-design occupancy, the opposite may also be true if significant portions of unoccupied indoor space are allowed to cool below comfortable temperature levels. Determining whether or not this is the case will require new large-scale datasets relating building occupancy and energy use. Further uncertainty analysis of building parameters (i.e., range of heating/cooling set-points, max hot water temperatures etc.) may also prove informative in this regard, however without empirical data on both the most important parameters (each building model contains many thousands of parameters) and the typical ranges encountered for the set of city building types modelled, it is unclear how informative this may be. Accordingly, collating data on the most influential parameters and their corresponding ranges for different building types would be a useful exercise. Our analysis also highlights how the different occupancy patterns and their assumed relationship with building loads implies different effectiveness for building efficiency measures. This is of particular importance, since energy efficiency incentives are vital in the real world of sustainable urban planning with limited budgets.

Through the inclusion of heterogeneous mobile phone inferred occupancy patterns, this work represents a significant improvement on current best practice occupancy estimates and an important step towards bespoke urban building energy models. It also has its set of limitations, including the lack of available real occupancy data on a sufficiently large scale for comparison. Therefore, we expect that as higher resolution data relating to urban mobility, building occupancy and building energy use become available, the presented modeling framework may be refined.

Finally, while our work has focused on improving urban building energy models with the mobile-inferred occupancy profiles, our results are also relevant for building utilization. The large discrepancy between estimated building occupancy and actual capacity implies that building efficiency may also be improved through better space utilization, for example by transitioning residential spaces into commercial spaces at times of low-residential occupancy. This is an interesting area of future urban planning and is especially important due to the high predicted expansion rates of global urban centers, although it has associated legislative and social barriers.

## Methods

**Mobile phone data and TimeGeo.** The TimeGeo framework[33] starts with the CDRs of 1.92 million people in the Greater Boston Area. The data were collected by AirSage for operational purposes for two mobile phone carriers. The location coordinates are estimated by the data provider using standard triangulation algorithms and have higher resolution (accuracy of 200–300 meters) compared to tower-based CDRs[55].

TimeGeo extracts stay points from each individual's sequence of consecutive mobile phone records, and separates commuters from non-commuters by identifying users with work stay points. For each tract in the Boston metropolitan area expansion factors are calculated for commuters and non-commuters using census data. Based on the distributions of empirical individual mobility parameters extracted from the active user data (0.78 million active users with homes inside the census region—Fig. 1a), the entire urban population movement is simulated. See[33].

**Building data for occupancy model.** Building data are obtained from the city of Boston, for 82,542 buildings. The geographic extent of this data is shown in Fig. 1b. Buildings are classified into several tax classes including residential classes, commercial classes, an industrial class and a mixed residential-commercial class (see Supplementary Fig. 2 and Supplementary Note 2). For mixed-use residential-commercial buildings, if the building is multi-floor we assume the first floor is commercial and if single-floored then we assume the floor area is split equally between residential and commercial classes.

**Building opening hours and Per Capita Areas**. We collect all the stays in each census tract using outlines provided by the Massachusetts 2010 census (and the latitude/longitude for each stay). For each building, we use the centroid and collect all buildings in each tract. We calculate the building total floor area, assuming each floor has equal area and the number of floors is the greatest integer of (building height)/(floor height). We assume all floor heights are 10 feet. We also add a broad classifier marking each building as residential, commercial, and industrial, which we use for stay compatibility.

To get building capacity, we need further information and we rely on a mapping between the functional use of buildings and the PCA required for that use case—see Supplementary Notes 2–4 and Supplementary Tables 1 and 2. Unfortunately, the set of functional uses for which PCAs are available is not coincident with the Boston buildings classification, nor is a credible translation possible, except for residential uses. Therefore we obtain additional use information at high spatial resolution from digital maps. We use the Google Places API Service and query for Points of Interest (POIs) located throughout Boston in the proximity of each non-residential building.

We aggregate the results to a list of POIs near non-residential buildings, discounting duplicate information. The information associated with each POI is variable and can include category (i.e., restaurant, bar, bank etc), place name, location, address, hours of operation, ratings, and reviews amongst others. Not all information fields are always available. We then use the building outlines to find POIs contained within each building and subsequently assign opening hours and PCAs.

Residential buildings are always open. For each non-residential building, the opening hours is the superset of the opening hours for all contained POIs. If there is no opening hour information then we assign opening hours of 7am–9pm (commercial) or 9am–5pm (industrial).

To assign PCAs, we map the different POI categories to a functional use for which we have an estimate of PCA (see Supplementary Note 4). Residential buildings are assigned 40 $m^2person^{-1}$. For non-residential buildings, we first assign PCAs to the set of buildings containing POIs—assigning the highest PCA for all contained POIs. Second, for each building without a PCA assigned, a PCA is randomly selected from the distribution of non-residential PCAs in that tract (inferred by POIs). In this way, if there are a high number of restaurants in a tract, an unmarked building would have a relatively higher probability of being allocated the PCA of a restaurant.

**Creating EnergyPlus building templates**. EnergyPlus building models were created using the UMI framework, as developed in ref. [10]. Using this tool building footprints were imported into Rhinoceros 3D CAD and extruded to the height specified by the data. Constructions, systems, and load characteristics were assigned to buildings using an appropriate template from the DOE reference buildings dataset, and based on the use and age of the structure as reported in Boston's tax assessment dataset (see Supplementary Note 6). EnergyPlus takes hourly occupancy as an input, defined by a combination of a maximum occupant density per unit floor area and a fractional hourly schedule. Additionally, inputs for occupant-driven loads are also required (i.e., lighting, equipment and hot water), again specified by peak values per unit area and a fractional schedule.

**Manipulating EnergyPlus input files and running simulations**. All adjustments to the generated building models, including changes to the occupancy patterns and implementing energy efficiency measures, were made in Python using Eppy[56], a python package for scripting EnergyPlus input files. For the 1,266 modeled buildings EnergyPlus was run for one representative day for each season. Weather data for the simulations was extracted from Boston's Typical Meteorological Year (TMY) data climate file provided by the US DOE[57], and developed from weather information gathered at Boston's Logan Airport station (~3 miles from the simulated site). The TMY data define a typical year's weather in the region and are commonly used for annual energy simulations. However, since this data is typical rather than extreme, it is not suitable for designing building systems for worst-case scenarios. Hence, the energyplus weather files (EPW files) also contain designations for the most extreme and most typical weather-weeks in each season. Since we are interested in understanding the average effect of occupancy on energy consumption and we only have a single day of mobile-inferred occupancy, we select the mid-range day from each typical week of weather data by season for our simulations. Therefore, each building was simulated for the days 30-Jan (winter), 1-Apr (spring), 30-Jul (summer), 23-Oct (autumn) under all scenarios.

## Data availability

Data to run the whole analysis for a demonstration census tract are provided along with the code—see Code Availability. Boston buildings data are available from the city of Boston's data portal Analyze Boston (https://data.boston.gov/). Larger samples of the data are available from the corresponding author upon reasonable request.

## Code availability

Python scripts to run the analysis are available at https://github.com/humnetlab/Mob_Building_Occ_Energy_Use.git, with stay point and building data provided for a demonstration census tract. This includes the source code used to generate the analysis

Figures in the manuscript and supplementary material. EnergyPlus (also required) is freely available from the US DOE (https://energyplus.net/). The UMI urban modelling interface used to create the energyplus building models is available from the sustainable design lab at MIT (http://web.mit.edu/sustainabledesignlab/projects/umi/index.html).

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

## Acknowledgements
This work was supported by grants from the Centre for Complex Engineering Systems at MIT-KACST, the MIT Energy Initiative and Philips Lighting.

## Author contributions
E.B., M.G., S.G. and J.K. conceived the research. E.B. and S.G. wrote the code for the occupancy analysis. C.C.D. and C.R. constructed the urban energy model. E.B., C.C.D. and C.R. developed the occupancy scenarios. E.B. performed the energy analysis. M.C.G., J.K. and C.R. supervised the research and provided guidance. E.B., C.C.D., C.R. and M.C.G. wrote the paper. All authors approved the manuscript.

## Additional information

**Competing interests:** The authors declare no competing interests.

