## [Peer Review File · Nature Communications]

Reviewers' comments:

Reviewer #1 (Remarks to the Author):

Study Overview

This study addresses an important and timely topic. The authors provide a framework to model Energy efficiency measures (retrofits) for buildings via physics-based models. These models require many detailed inputs in order to produce accurate results. Therefore, the authors embarked on a two-part study where they first developed an estimated building occupancy from mobile phone data and then used DOE (Department of Energy) EnergyPlus simulation engine to model 1224 buildings in Boston. Overall, the study concluded that the Reference Building models developed by the National Renewable Energy Laboratory (NREL) overestimated the energy consumption by 2-17% for the studied stock of 1224 buildings in Boston.

Review comments

(1) Page 2, Line 12 --- “previous energy models ONLY considered single buildings”

This statement is incorrect. Please, see the following paper:

---Reference [8] in the paper: Yixing Chen, Tianzhen Hong, Mary AnnPiette, Automatic generation and simulation of urban building energy models based on city datasets for city-scale building retrofit analysis, Applied Energy, 205:323–335, 2017.

This existing paper actually applied the same methodology: EnergyPlus simulation engine with NREL Reference Building models, and building property database maintained by LBNL (Lawrence Berkeley National Laboratory). How is the present study different methodologically from the perspective of energy modeling? The assessment of occupancy defined improvements compared to the current modeling methods, but the energy modeling uses an existing approach.

(2) Page 2, Line 12 ---- “or reduced statistical models [29].”

Statistical modelers were the first ones to look at energy efficiency measures for sustainable cities. Why would the current study overlook statistical models of city energy efficiency measure analyses? In fact, statistical modelers were the first ones to push analyses of building energy efficiency measures for an entire city. The modeling approaches that use large databases to identify most

promising energy efficiency measures was pioneered in these studies (please, notice that this is just a sample of studies and they are all published in reputable journals):

---Abdolhosseini Qomi MJ, Noshadravan A, Sobstyl JM, Toole J, Ferreira J, Pellenq RJ-M, Ulm F-J, Gonzalez MC. 2016 Data analytics for simplifying thermal efficiency planning in cities. J. R. Soc. Interface 13: 20150971.

---Constantine E. Kontokosta , Christopher Tull. A data-driven predictive model of city-scale energy use in buildings Applied Energy 197 (2017) 303–317

(3) Page 2, Line 14 – “(UBEMs) have been developed [31, 15, 45, 21, 28], allowing the simulation of thousands”

It seems that the authors missed a large body of work done by researchers in Switzerland and UK (this is just a sample of relevant publications in reputable journals):

----A review on occupant behavior in urban building energy models Gabriel Happle, Jimeno A. Fonseca and Arno Schlueter Energy & Buildings, vol. 174, pp. 276-292, Amsterdam: Elsevier, 2018

----An open-source simulation platform to support the formulation of housing stock decarbonisation strategies G Sousa, BM Jones, PA Mirzaei, D Robinson, Energy and Buildings 172, 459-477

----A generalised stochastic model for the simulation of occupant presence J Page, D Robinson, N Morel, JL Scartezini Energy and buildings 40 (2), 83-98 506 2008

(4) Page 3, line 26 to 34 ----“demonstrating that sparse mobile phone data can be used to model individual trajectories for entire urban populations”

TimeGeo model is a nice development and an obviously better approach than the currently used methods. It would be useful to briefly state not just the advantages of TimeGeo here, but also the limitations of the method when compared to other reviewed methods.

(5) Page 6, Line 110 -- “type raised to the power 3/2.”

Please, specify how and why this particular assumption was made. This seems to be an arbitrary choice without an explanation.

(6) Page 8, Line 163 ---“Since occupancy is often a large driver for building energy use...”

This statement contradicts the cited study [25]. In fact, study [25] concludes that occupancy was not the primary driver for building energy consumption. Here is an excerpt from reference [25] abstract: “The results of the study demonstrate: the operation of the heating, ventilation and air conditioning (HVAC) systems adhered more closely to factors other than occupancy i.e. external temperature, whilst a small part of the electricity levels did correlate with the occupancy.”

(7) Page 10, Line 213-214 ---“We simulate one representative day in each season (see Materials and Methods) and present results averaged across all seasons.

Why is the study reducing the analysis to a single day in each season and presenting only averaged results across all seasons? This appears to be a temporal reduction in analysis that decreases importance of refined occupancy data and use of physic-based models.

(8) Page 10, Line 242-243 --- “by 10% in comparison to the 2010 building code requirements for all residential space, and secondly, we implement a 10% efficiency gain for equipment in commercial”

It is unclear why these two particular energy efficiency measures are selected among all different options. This selection appears to be arbitrary. Please, explain.

(9) Page 11, Line 281 – “we find that total building energy use is predicted 2–17% lower when the mobile occupancy”

The current study solely compares Energy Plus results for Reference Buildings with Energy Plus results created by the authors (high and low scenarios), so the study conclusions are solely based on numerical data from physics-based models that might be inaccurate. How accurate are these energy modeling results? How was that energy modeling accuracy assessed?

(10) Page 12, line 307- 308 “It also has its own set of limitations, including the lack of available real occupancy data on a sufficiently large scale for comparison”

The accuracy of occupancy predictions is an issue that is already discussed in this excerpt from the paper. However, a bigger issue is that occupancy results are practically averaged into two scenarios (fractional hourly schedules). Therefore, the way that the study was structured and executed, it

ended up being a massive sensitivity analysis using EnergyPlus simulation engine with averaged data from TimeGeo. Therefore, the presented conclusions on energy use intensities are primarily order of magnitude estimates. Please, explain why this approach is appropriate or advantageous compared to the existing state-of-the-art.

Reviewer #2 (Remarks to the Author):

The paper presented an interesting study using large-scale mobile phone data, analysis and urban building energy modeling to find out actual occupancy in commercial buildings are much lower than current engineering practice (used in DOE reference building models) and discuss the impact on building energy use. The study is well structured and the paper is well written. The reviewer has the following concerns and suggestions that should be addressed before publication.

1. The study should describe more details about the mobile phone data. In the Boston city area, there are multiple mobile phone service providers, does the data cover the entire services or only one specific phone company? How the phone data is anonymized to protect privacy (so no information can be derived from one home to a workplace for that specific people/phone)? Are such phone data always available and easy to get for research purpose to cover broad areas and longer time period? Or this is just one-off study? From real application point of view, is this feasible and affordable?

2. Occupancy is an important and varying factor in building design and performance, as the authors pointed out. The static (not stochastic) occupancy information used in the DOE reference buildings are not intended to represent actual operation situation or diversity between buildings. They are used for design day peak load calculation as well as annual energy consumption estimation. They represent some sort of full occupancy rate in order to size the HVAC equipment to ensure adequate capacity for various uses in the building life cycle. So sometimes occupancy rate may be 50% while it can get as high as 90% especially during some seasonal events. The authors' conclusion that the estimated actual occupancy is only 20 something percent of the design occupancy can mislead people. The mobile phone data is only 6 weeks, is it adequate to capture seasonal variations (which is strong for commercial buildings like hotels)?

3. Uncertainty quantification is missing. What is the accuracy of using mobile connectivity to predict occupancy? Authors recognized there lacks ground truth occupancy data to validate the results.

4. The statement "The EPW file defines a typical week of weather data by season" is not accurate. The EnergyPlus epw weather file (TMY3) usually contains 8760 hourly (entire year) weather data.

5. Evaluating the impact of occupancy on building energy use depends upon their actual energy systems and their control - whether there are occupancy-driven controls. Authors employed a rational approach to capture two bounds - one less responsive to occupancy while the other more (e.g. with demand-control ventilation, occupancy-responsive lighting and plug-loads).

6. In urban scale building energy modeling, usually certain techniques are employed to calibrate the models to ensure reliable results. For example, using tract-level actual building energy consumption data to tune key model parameters which include the occupancy rate, operating schedules, internal heat gains, HVAC system efficiency. Occupancy is one of the factors but others can be as important.

7. The DOE reference building models are developed for modeling individual building energy use to support evaluation of energy conservation measures or efficient design. They do not capture realistic diversity between buildings or actual operation conditions. They are not intended for use in a homogeneous way to predict urban scale building energy use. Prototype cities or districts energy models will be more appropriate for such use cases.

Reviewer #3 (Remarks to the Author):

While this paper provides an interesting read and works to address a largely under-researched topic using a novel approach, I have several concerns regarding the underlying assumptions used in their model. Without validating these assumptions or the results, or performing a thorough uncertainty analysis, I worry that the significance and magnitude of the results may not accurately represent the Boston urban energy landscape and more generally urban building energy.

1. In SI line 51, the authors indicate the large variation in PCA values for buildings of the same type. They use the DOE Commercial Reference Buildings; however, that includes only 16 building types. For all other commercial buildings, they use building standards and codes. The use of buildings standards and codes to generate the underlying assumptions of your model seems to conflict with one of the major claims of the paper as shown in the abstract: "Our results demonstrate that, on the urban scale, occupancy differs significantly from widely used standards-based assumptions."

2. Table 2 in the SI shows how points of interest are mapped to building types. There appears to be errors in this table. For example, spa, beauty salon, and haircare are all mapped to athletic facilities, while a stadium is mapped to a place of worship.

3. In section 4 of the SI, you recognize that expert energy modelers are used to model individual building energy use but explain that for the purpose of this paper a streamlined, automated approach was taken and a new tool developed using average building characteristics (commonly assigned through “archetypes”). Has this tool been validated against the results from expert energy modelers? Other options of validation that you may want to consider are the EIA’s Commercial Building Energy Consumption Survey and Residential Energy Consumption Survey. Another approach would be to look at the energy consumption data from utilities or energy authorities.

4. I’m a bit confused by which buildings are actually included in your analysis. When reading the main text, I thought all buildings were being classified and included. Then in the SI, it reads “we have chosen to only model the energy use of buildings falling within the classes available in the DOE dataset.” What fraction of the actual buildings in the Back Bay neighborhood does this represent? How do these buildings differ from those that are not included in the study? Does this elimination of buildings in the analysis bias the results and, if so, by how much?

5. I’m a bit confused by your low and high impact calculations. These are defined by absolute and relative mobile occupancy. However, you only have data on 20% of the cell phone users in the area. I assume you are using TimeGeo to scale this data for the rest of the population and that these scaled values is what actually is going into Eqns 2-3. If this is the case, please clarify this in the text surrounding these equations. Additionally, if this is the case, your scaling will introduce uncertainty and this is not discussed. This should be quantified. The uncertainty in the model will influence the significance of the results.

6. Line 76: The mobile data is collected for a 6 week period in 2010. When were these 6 weeks? Were they consecutive weeks or did they include different seasons? Did they include holidays, such as Christmas, that would alter typical patterns of shopping? How significant was the variation found between weeks and between seasons?

7. Lines 83-84: You assume the people only visit buildings within their own census tract. According to the Boston Globe (<https://www.bostonglobe.com/metro/2018/09/05/mass-average-commute-time-increases-longest-country/jt4VNcKMleeY1MGaxl3XYO/story.html>), Boston has the fifth longest commute to work in the country. Using the data that you have, you can test the validity of this assumption. In your set of active phone users, what fraction of users never left their census tract in

the 6 week period? I would suggest reporting this fraction in the text to validate this assumption, or, depending on the result, modifying this assumption. This is one of your assumptions that I think may radically alter your results. You mention in lines 144-146 that the mobile occupancy data differed by the DOE reference model by an order of magnitude at some times. I wonder if this discrepancy occurs since you only consider individuals staying in their residential census tract and miss all those who commute into the city for work.

8. I had a difficult time understanding Eqn. 1. What is the justification for raising the relative capacity to a factor of $3/2$? Has this attraction factor been validated? What is the uncertainty in this factor?

Other comments:

9. Line 1: The use of the word “huge” is vague. Provide a more quantitative measure.

10. Figure 1: I’m not sure what the black boxes in A and B represent. Is B the blow up of the black box in A, and C the blow up of the black box in B? The expansion factors, as indicated by color, in A cannot be discerned for the census tracts within the black box.

Overall, I worry that assumption and model uncertainty and bias have not been well documented throughout the paper, making it difficult to interpret the validity of the results. I would strongly advise quantifying this uncertainty and bias prior to publication.

Detailed Response to review comments

The reviewer comments are italicized and our responses are indented and highlighted in blue

Reviewer #1 (Remarks to the Author):

Study Overview

This study addresses an important and timely topic. The authors provide a framework to model Energy efficiency measures (retrofits) for buildings via physics-based models. These models require many detailed inputs in order to produce accurate results. Therefore, the authors embarked on a two-part study where they first developed an estimated building occupancy from mobile phone data and then used DOE (Department of Energy) EnergyPlus simulation engine to model 1224 buildings in Boston. Overall, the study concluded that the Reference Building models developed by the National Renewable Energy Laboratory (NREL) overestimated the energy consumption by 2-17% for the studied stock of 1224 buildings in Boston.

We thank the reviewer for pointing out the importance of our paper's subject.

Review comments

(1) Page 2, Line 12 --- "previous energy models ONLY considered single buildings"

This statement is incorrect. Please, see the following paper:

---Reference [8] in the paper: Yixing Chen, Tianzhen Hong, Mary AnnPiette, Automatic generation and simulation of urban building energy models based on city datasets for city-scale building retrofit analysis, Applied Energy, 205:323–335, 2017.

This existing paper actually applied the same methodology: EnergyPlus simulation engine with NREL Reference Building models, and building property database maintained by LBNL (Lawrence Berkeley National Laboratory). How is the present study different methodologically from the perspective of energy modeling? The assessment of occupancy defined improvements compared to the current modeling methods, but the energy modeling uses an existing approach.

We thank the reviewer for pointing out this mistake. The sentence was intended to convey that urban-scale building energy models are a recent development, however we agree that the sentence is misleading. We have clarified this in the re-written introduction.

We indeed use a very similar methodology from the perspective of energy modelling as the referenced paper [A], rather building on the framework from [B] (which again is similar and uses energyplus as the simulation engine along with building archetypes based on the NREL Reference models and the construction year). Therefore, the primary contribution of this work is the assessment of occupancy on an urban scale and the resulting implications for urban building energy modelling. As highlighted by the reviewer, recent comprehensive reviews of occupancy for urban building energy models

[C, D] highlights that the lack of understanding regarding occupant presence in buildings at the urban scale is a major issue for urban energy policy formation.

[A] Yixing Chen, Tianzhen Hong, Mary AnnPiette, Automatic generation and simulation of urban building energy models based on city datasets for city-scale building retrofit analysis, *Applied Energy*, 205:323–335, 2017.

[B] Davila, Carlos Cerezo, Christoph F. Reinhart, and Jamie L. Bemis. "Modeling Boston: A workflow for the efficient generation and maintenance of urban building energy models from existing geospatial datasets." *Energy* 117 (2016): 237-250.

[C] A review on occupant behavior in urban building energy models Gabriel Happle, Jimeno A. Fonseca and Arno Schlueter *Energy & Buildings*, vol. 174, pp. 276-292, Amsterdam: Elsevier, 2018

[D] Zhang, Y., Bai, X., Mills, F. P., & Pezzey, J. C. (2018). Rethinking the role of occupant behavior in building energy performance: A review. *Energy and Buildings*.

(2) Page 2, Line 12 ---- “or reduced statistical models [29].”

Statistical modelers were the first ones to look at energy efficiency measures for sustainable cities. Why would the current study overlook statistical models of city energy efficiency measure analyses? In fact, statistical modelers were the first ones to push analyses of building energy efficiency measures for an entire city. The modeling approaches that use large databases to identify most promising energy efficiency measures was pioneered in these studies (please, notice that this is just a sample of studies and they are all published in reputable journals):

---Abdolhosseini Qomi MJ, Noshadravan A, Sobstyl JM, Toole J, Ferreira J, Pellenq RJ-M, Ulm F-J, Gonzalez MC. 2016 *Data analytics for simplifying thermal efficiency planning in cities. J. R. Soc. Interface* 13: 20150971.

---Constantine E. Kontokosta , Christopher Tull. *A data-driven predictive model of city-scale energy use in buildings Applied Energy* 197 (2017) 303–317

We agree with the reviewer that we have understated the relevance and importance of statistical models and have expanded our description of these type of urban energy models, ensuring to include the suggested references. We note that these statistical models are particularly useful for benchmarking building energy performance and were the first to evaluate the urban-scale impacts of certain Energy Conservation Measures (ECMs).

We also highlight that existing statistical models are limited by the explanatory variables selected and therefore can suffer from poor out of sample predictions relating to ECMs, due to, for example, feedback between building operation and occupants behavior. Moreover, it would be very difficult to obtain information about operations, occupancy, and existing conservation measures with reasonable sample sizes. Therefore, the versatility of bottom-up engineering UBEMs is important to allow planners to quantitatively assess retrofit strategies and energy supply options [A]. Furthermore,

bottom-up UBEMs are useful for designing high-efficiency future urban areas, which will be required to house future booming city populations.

[A] A review on occupant behavior in urban building energy models Gabriel Happle, Jimeno A. Fonseca and Arno Schlueter *Energy & Buildings*, vol. 174, pp. 276-292, Amsterdam: Elsevier, 2018

(3) Page 2, Line 14 – “(UBEMs) have been developed [31, 15, 45, 21, 28], allowing the simulation of thousands”

It seems that the authors missed a large body of work done by researchers in Switzerland and UK (this is just a sample of relevant publications in reputable journals):

-----A review on occupant behavior in urban building energy models Gabriel Happle, Jimeno A. Fonseca and Arno Schlueter *Energy & Buildings*, vol. 174, pp. 276-292, Amsterdam: Elsevier, 2018

-----An open-source simulation platform to support the formulation of housing stock decarbonisation strategies G Sousa, BM Jones, PA Mirzaei, D Robinson, *Energy and Buildings* 172, 459-477

-----A generalised stochastic model for the simulation of occupant presence J Page, D Robinson, N Morel, JL Scartezzini *Energy and buildings* 40 (2), 83-98 506 2008

We are very grateful to the reviewer for bringing this important work to our attention. The suggested references have been included and we have extended the introductory section of the paper, adding a paragraph highlighting the importance of understanding building occupancy for energy modeling. We have included the suggested references here and are open to include any previous work that highlights the importance of urban building occupancy for urban building energy models.

We also agree with the reviewer that models exist for translating occupancy to electricity and heat loads in individual buildings. However, no such models exist for urban-scale simulations and therefore the DOE reference buildings represent the only current option for urban scale models, as is emphasized by the first of the suggested works highlighted by the reviewer. Furthermore, all currently existing models for individual buildings require empirical knowledge of the maximum building occupant presence, which clearly is difficult to obtain for urban-scale work.

(4) Page 3, line 26 to 34 ----“demonstrating that sparse mobile phone data can be used to model individual trajectories for entire urban populations”

TimeGeo model is a nice development and an obviously better approach than the currently used methods. It would be useful to briefly state not just the advantages of TimeGeo here, but also the limitations of the method when compared to other reviewed methods.

We have extended the section “From mobile phone stay points to building occupancy” to include a discussion of the advantages and limitations of the TimeGeo model. We thank the referee for the valuable recommendation.

(5) Page 6, Line 110 -- “type raised to the power $3/2$.”

Please, specify how and why this particular assumption was made. This seems to be an arbitrary choice without an explanation.

We agree with the reviewer that insufficient detail regarding both choice and introduction of this parameter (now denoted μ) was given in the first version of the manuscript. Therefore, in the revised manuscript we provide more clarity and a sensitivity analysis to the value of this parameter. Our intuition for introducing the attraction factor is to mimic the rich-get-richer effect in the popularity of places, *i.e.* we anticipate that areas with more capacity for occupants performing a particular type of activity will attract more people per unit area than those areas with less opportunity for that activity type. The mobile phone data already has this effect at the census tract scale, since we observe that census tracts with large numbers of shops and restaurants attract more ‘other type’ stays. The parameter μ creates this effect on a city block scale within the census tracts and as a result, we get clustering of occupants where there are clusters of similar facilities. This leads to popular districts for certain types of activities – *i.e.* shopping or restaurant districts.

We included a study of the sensitivity of the resulting occupancy to the exact value of this exponent. Expressing the exponent as $(1+\mu)$ in the revised manuscript, we perform a sensitivity analysis in the range μ is 0-1. We believe this is a reasonable range since if $\mu < 0$ then districts with lots of opportunities will be less popular per unit floor space and $\mu > 1$ leads to unreasonably large occupancies in certain buildings. Details of the sensitivity analysis are provided in the main text and Supplementary Note 5. Although larger μ shifts the occupancy distribution for the buildings towards a few more highly occupied spaces, the effect of this parameter is small in comparison to the differences between the mobile-inferred and DOE reference occupancy.

(6) Page 8, Line 163 ---“Since occupancy is often a large driver for building energy use...”

This statement contradicts the cited study [25]. In fact, study [25] concludes that occupancy was not the primary driver for building energy consumption. Here is an excerpt from reference [25] abstract: “The results of the study demonstrate: the operation of the heating, ventilation and air conditioning (HVAC) systems adhered more closely to factors other than occupancy *i.e.* external temperature, whilst a small part of the electricity levels did correlate with the occupancy.”

It is certainly true that for a minority of buildings, energy use is mostly independent of occupancy. The study [25] considers two buildings on the MIT campus and therefore cannot be considered as generally representative of a large number of buildings. Notably, one of the buildings contained laboratories with equipment that was run

independent of human presence. Indeed of the more 'office-like' building, nearly 70% of the variation in electricity consumption could be accounted for by changing occupancy.

We agree that using this reference in this context is highly misleading and it has been removed.

(7) Page 10, Line 213-214 ---“We simulate one representative day in each season (see Materials and Methods) and present results averaged across all seasons.

Why is the study reducing the analysis to a single day in each season and presenting only averaged results across all seasons? This appears to be a temporal reduction in analysis that decreases importance of refined occupancy data and use of physic-based models.

We utilize typical days in each season since TimeGeo produces results for a single typical weekday which was validated by mainstream sources. Our aim in this study is to understand how typical building occupancies compare between the mobile inferred-model and the DOE reference building models. Therefore, since we are interested in average effects, rather than extreme effects, we believe that the results of the comparison are valid. Certainly, more mobile phone data would allow us to produce different day types and reproduce seasonal effects. However, given the huge discrepancy between the *typical* mobile-inferred occupancy and that of the reference buildings, our study certainly highlights that in general for most buildings, occupancy is likely to be significantly different to standard assumptions currently used in urban building energy models.

(8) Page 10, Line 242-243 --- “by 10% in comparison to the 2010 building code requirements for all residential space, and secondly, we implement a 10% efficiency gain for equipment in commercial”

It is unclear why these two particular energy efficiency measures are selected among all different options. This selection appears to be arbitrary. Please, explain.

These two generic energy efficiency measures have been chosen as potential measures which could be broadly promoted through energy policy, representing two *contrasting* efficiency mechanisms (improved insulation vs improved equipment efficiency). The purpose of the paper is to illustrate that occupancy may have differing effects on different energy efficiency measures. Indeed our results suggest that is the case, given that using our occupancy estimates compared to those in the DOE reference buildings leads to an increased benefit from insulation increases and a lower benefit from equipment upgrades.

(9) Page 11, Line 281 – “we find that total building energy use is predicted 2–17% lower when the mobile occupancy”

The current study solely compares Energy Plus results for Reference Buildings with Energy Plus results created by the authors (high and low scenarios), so the study conclusions are solely based on numerical data from physics-based models that might be inaccurate. How accurate are these energy modeling results? How was that energy modeling accuracy assessed?

Since the energy data in our modelled region is unavailable at sufficient resolution for any useful comparison, we cannot explicitly test the accuracy of the energy modeling. However, the underlying reference model (the DOE reference scenario) has been cross-checked against the national average EUIs for the equivalent building types in the Commercial Building Energy Consumption Survey and the Residential Building Energy Consumption Survey. The average difference of the modeled EUI compared to the CBECS/RBECS averages ranged between 5% and 20% for the modeled building types with the simulated value always being higher than the CBECS reference [A].

The purpose of our study is to highlight that current standards for occupancy in city-scale building energy models are highly inaccurate and that mobile phone data can help to overcome this challenge. These models are used both in designing at a city-block level and to understand the effectiveness of urban energy policy, and furthermore, they represent the state-of-the-art in neighborhood energy assessments for both planning and policy-design. We demonstrate that these can be improved using passive model phone data at urban scale.

[A] Carlos Cerezo Davila, Christoph F Reinhart, and Jamie L Bemis. Modeling boston: A workflow for the efficient generation and maintenance of urban building energy models from existing geospatial datasets. *Energy*, 117:237–250, 2016.

(10) Page 12, line 307- 308 “It also has its own set of limitations, including the lack of available real occupancy data on a sufficiently large scale for comparison”

The accuracy of occupancy predictions is an issue that is already discussed in this excerpt from the paper. However, a bigger issue is that occupancy results are practically averaged into two scenarios (fractional hourly schedules). Therefore, the way that the study was structured and executed, it ended up being a massive sensitivity analysis using EnergyPlus simulation engine with averaged data from TimeGeo. Therefore, the presented conclusions on energy use intensities are primarily order of magnitude estimates. Please, explain why this approach is appropriate or advantageous compared to the existing state-of-the-art.

In urban energy modelling at a neighborhood or city scale, the state-of-the-art is to generate building physics models (based around energyplus or some other detailed building simulation engine) based on archetypes. These archetypes contain information regarding building occupancy. We explicitly demonstrate that the occupancy used in these state-of-the-art models is likely to be severely inaccurate at a neighborhood scale. Our work demonstrates (1) that mobile phone data can be used to improve these state-of-the-art energy models (2) that differing occupancy implies different effectiveness for

ECMs and is therefore important for energy-policy design, and (3) our work highlights that further work relating occupancy to occupant-driven-loads on this scale is essential for improving urban energy models.

Therefore, we agree that the two occupancy-scenarios could be improved with more empirical data, however we also consider them to be a rational approach for the two bounds without any available model or data that functions on this scale. Essentially, these two scenarios can be considered as one less responsive to occupancy while the other more (i.e. if ventilation, lighting and plug-loads were all controlled by occupancy). Therefore, one important aspiration of this work is to motivate further studies to uncover data which can be used to better relate improved estimates of occupant presence (like our study) with behavior models that have been developed on an urban scale.

Reviewer #2 (Remarks to the Author):

The paper presented an interesting study using large-scale mobile phone data, analysis and urban building energy modeling to find out actual occupancy in commercial buildings are much lower than current engineering practice (used in DOE reference building models) and discuss the impact on building energy use. The study is well structured and the paper is well written. The reviewer has the following concerns and suggestions that should be addressed before publication.

We thank the reviewer for their positive remarks.

1. The study should describe more details about the mobile phone data. In the Boston city area, there are multiple mobile phone service providers, does the data cover the entire services or only one specific phone company? How the phone data is anonymized to protect privacy (so no information can be derived from one home to a workplace for that specific people/phone)? Are such phone data always available and easy to get for research purpose to cover broad areas and longer time period? Or this is just one-off study? From real application point of view, is this feasible and affordable?

We have added more details about the mobile phone data in the introductory section of the paper and the section entitled "From mobile phone stay points to building occupancy".

The mobile phone data that we use to develop the TimeGeo model is anonymized by giving each consumer a randomized id. However, the reviewer is right that identities could likely be deduced through identifying home and work locations. The TimeGeo model however, uses the distribution of the observed mobility patterns to simulate the movements of the entire Boston metro population. Therefore, we use 3.5 million simulated users, rather than real users in this paper. The journeys for the 3.5 million simulated users align with best-in-class transportation demand models ([A] and

Supplementary Note 1) and are considered to be statistically identical to those of the real users at the census tract level.

The methodology we develop is portable to any location where the data is available. The required data consists of mobile phone data, building outlines and heights, building classification and places available in digital maps to imply a functional use. We believe these sources to be sufficiently ubiquitous in many different locations.

[A] Jiang, S., Yang, Y., Gupta, S., Veneziano, D., Athavale, S., & González, M. C. (2016). The TimeGeo modeling framework for urban mobility without travel surveys. *Proceedings of the National Academy of Sciences*, 113(37), E5370-E5378.

2. Occupancy is an important and varying factor in building design and performance, as the authors pointed out. The static (not stochastic) occupancy information used in the DOE reference buildings are not intended to represent actual operation situation or diversity between buildings. They are used for design day peak load calculation as well as annual energy consumption estimation. They represent some sort of full occupancy rate in order to size the HVAC equipment to ensure adequate capacity for various uses in the building life cycle. So sometimes occupancy rate may be 50% while it can get as high as 90% especially during some seasonal events. The authors' conclusion that the estimated actual occupancy is only 20 something percent of the design occupancy can mislead people. The mobile phone data is only 6 weeks, is it adequate to capture seasonal variations (which is strong for commercial buildings like hotels)?

We agree with the reviewer that the DOE reference buildings are used for design day peak load calculations and that occupancy on the peak-load day may indeed be a large fraction of the designated peak occupancy. However, the reference buildings also represent the state-of-the-art for designing both energy-policy related to Energy Conservation Measures (ECMs) and new urban neighborhoods, for which understanding the real *average or typical* occupancy is crucial.

Therefore, our work is not intended to improve peak-load day calculations for individual buildings (nor even for neighborhoods since the peak-load days for many buildings may be highly correlated) but is intended to improve the estimates of average building performance. This average performance is equally as important for both building operators and system operators to know in order to minimize a building or neighborhood's environmental footprint.

Furthermore, we note that Mobile phone data frameworks based on average days can act as a starting point and has been adapted to model mega traffic events [A].

These points have been clarified in the manuscript.

[A] Xu, Yanyan, and Marta C. González. "Collective benefits in traffic during mega events via the use of information technologies." *Journal of The Royal Society Interface* 14.129 (2017): 20161041.

3. Uncertainty quantification is missing. What is the accuracy of using mobile connectivity to predict occupancy? Authors recognized there lacks ground truth occupancy data to validate the results.

We agree with the reviewer that the paper could be improved through further justification of the mobile phone based occupancy, however, this is currently unavailable. Therefore, we have emphasized the reasons why mobile phone data indeed appears to offer a more reasonable estimate of building occupancy on an urban scale. Fig. 2B, added in the revised manuscript, illustrates that the mobile phone data seems much more reasonable considering the census population of the modelled district.

Furthermore, we have introduced a sensitivity analysis regarding the effect of the attraction factor μ in Equation 1 in the revised manuscript on building occupancy. We find that within a reasonable range of values, the change introduced by μ is small compared to the difference between the mobile inferred and reference building occupancies. This is demonstrated by supplementary Figure 7 as shown below and included in supplementary note 5.

Therefore, while the exact building occupancy lacks ground truth (no data sources are available that could be used to the best of the authors knowledge at the required scale), we see that occupancy which is consistent with travel demand models and the census necessarily implies a large deviation with the reference building occupancy with a very high degree of confidence.

Figure 7: Median daily occupancy fraction at different μ values. (A) Residential. (B) Commercial. The reference case value is also shown.

4. The statement "The EPW file defines a typical week of weather data by season" is not accurate. The EnergyPlus epw weather file (TMY3) usually contains 8760 hourly (entire year) weather data.

Weather data for the simulations was extracted from Boston's Typical Meteorological Year (TMY) data climate file provided by the US DOE, and developed from weather information gathered at Boston's Logan Airport station (approximately 3 miles from the simulated site). The TMY data defines a typical year's weather in the region and is commonly used for annual energy simulations.

Within the epw weather file, to help with creating design days that either represent extreme cases or more typical periods for each season, certain weeks are highlighted in each season as either *Week Nearest Average Temperature For Period* or *Week Nearest Min/Max Temperature For Period*.

Since we are interested in understanding the average effect of occupancy on energy use and we only have a single day of typical occupancy, we select one day from each typical week of weather data by season for our simulations.

5. Evaluating the impact of occupancy on building energy use depends upon their actual energy systems and their control - whether there are occupancy-driven controls. Authors employed a rational approach to capture two bounds - one less responsive to occupancy while the other more (e.g. with demand-control ventilation, occupancy-responsive lighting and plug-loads).

We thank the reviewer for this comment. The two scenarios are indeed intended to rationally represent the two bounds of responsiveness to occupancy, one less responsive to occupancy while the other more.

6. In urban scale building energy modeling, usually certain techniques are employed to calibrate the models to ensure reliable results. For example, using tract-level actual building energy consumption data to tune key model parameters which include the occupancy rate, operating schedules, internal heat gains, HVAC system efficiency. Occupancy is one of the factors but others can be as important.

We have clarified the manuscript to emphasize that our baseline model is based on the state-of-the-art in urban energy modelling. This baseline model (developed in [A]) calibrates the predicted energy consumption against available US building energy consumption surveys and ensures that it is consistent with large scale energy consumption datasets for the region [A]. The purpose of this paper is to highlight how the occupancy can be improved in these models and to highlight the importance that this has on designing policy related to urban ECMs.

[A] Carlos Cerezo Davila, Christoph F Reinhart, and Jamie L Bemis. Modeling boston: A workflow for the efficient generation and maintenance of urban building energy models from existing geospatial datasets. *Energy*, 117:237–250, 2016.

7. The DOE reference building models are developed for modeling individual building energy use to support evaluation of energy conservation measures or efficient design. They do not capture realistic diversity between buildings or actual operation conditions. They are not intended for use in a homogeneous way to predict urban scale building energy use. Prototype cities or districts energy models will be more appropriate for such use cases.

In general, two types of city scale energy models exist - those that are either statistical models based on data or those that are based on a detailed building-physics simulation engine. Although improving as more data becomes available, the primary limitations of the first model class are (1) the lack of detail regarding individual buildings which can lead to poor out-of-sample predictions and (2) they cannot be used to design new urban neighborhoods.

The state-of-the-art in prototype cities or district energy models are building simulations which include detailed physics-based simulations of thousands of individual buildings (i.e. [A,B,C]). These prototype energy models capture the diversity of individual buildings using highly automated workflows that assign templates to buildings based on building type and construction period. In the US, the state-of-the-art templates are the provided by the DOE reference building models. These models can then account for differences based on shading, geometry, orientation and constructions, however currently differences based on occupancy cannot be accounted for. Furthermore, this is an issue for all national building stock models.

Therefore, while the state-of-the-art city or district energy models are based on the DOE reference building datasets, we agree with the reviewer that diversity between different buildings is insufficiently captured with these methods. Indeed, our DOE reference scenario is a state-of-the-art prototype district energy model.

That is precisely the reason why we develop this method for modeling the *occupancy-driven diversity* in these state-of-the-art city energy models.

[A] Carlos Cerezo Davila, Christoph F Reinhart, and Jamie L Bemis. Modeling boston: A workflow for the efficient generation and maintenance of urban building energy models from existing geospatial datasets. *Energy*, 117:237–250, 2016.

[B] Hong, T., Chen, Y., Lee, S. H., & Piette, M. A. (2016). CityBES: A web-based platform to support city-scale building energy efficiency. *Urban Computing*.

[C] Kazas, G., Fabrizio, E., & Perino, M. (2017). Energy demand profile generation with detailed time resolution at an urban district scale: A reference building approach and case study. *Applied energy*, 193, 243-262.

Reviewer #3 (Remarks to the Author):

While this paper provides an interesting read and works to address a largely under-researched topic using a novel approach, I have several concerns regarding the underlying assumptions used in their model. Without validating these assumptions or the results, or performing a thorough uncertainty analysis, I worry that the significance and magnitude of the results may not accurately represent the Boston urban energy landscape and more generally urban building energy.

We would like to clarify the core contribution and novelty of this work. Statistical estimates of infrastructure demand from mobile phone data, as the one proposed here, have been successful in route assignments and traffic planning. This work proposes an statistical framework to bring this data to the service of Urban Building Energy Models via statistical estimates of buildings occupancy. We have expanded the introduction to TimeGeo, clarifying that it accounts for the entire urban population and the predicted movements are consistent with best-in-class urban mobility models as well as the census population. We discuss the limitations of the model which primarily relate to the resolution and duration of the data used. Further work of the urban informatics community, working in similar frameworks for human mobility modeling, can be attracted to the uses of digital traces for the UBEMs domain.

While the exact building occupancy lacks ground truth (no data sources are available that could be used to the best of the authors knowledge at the required scale), we see that occupancy which is consistent with travel demand models and the census necessarily implies a large deviation with the reference building occupancy with a very high degree of confidence. Furthermore, this large discrepancy is significant for energy policy planning, since UBEMs are crucial for understanding the effect of energy efficiency measures and no alternative to reference building occupancy profiles currently exists on an urban scale.

1. In SI line 51, the authors indicate the large variation in PCA values for buildings of the same type. They use the DOE Commercial Reference Buildings; however, that includes only 16 building types. For all other commercial buildings, they use building standards and codes. The use of buildings standards and codes to generate the underlying assumptions of your model seems to conflict with one of the major claims of the paper as shown in the abstract: “Our results demonstrate that, on the urban scale, occupancy differs significantly from widely used standards-based assumptions.”

In order to estimate the building occupancy, we first get the building type by using information in digital maps. Then, we consider that each building attracts people proportional to its *nominal capacity*. To calculate each building’s unique nominal capacity, we use typical PCA values for buildings of that type, which, where available, are taken from the DOE Commercial Reference Buildings (see Supplementary Note 2).

Therefore, it is true to say that, for lack of better data, the nominal building capacities are based on standards. For example, our model will consider that two buildings of the same type with the same floor area will have the same *nominal capacity*.

However, when assigning people to buildings, the *number of people* assigned will depend on how many mobile-phone based stay points are observed in the building's tract and, as demonstrated by Equations 1 and 2 in the revised manuscript, it also depends on the other buildings in the tract and their spatial distribution. To illustrate, consider two identical buildings in different tracts, one for which the empirical mobile phone data implies there are many more stay points in the surrounding vicinity. Although these buildings will have the same *nominal capacity*, the estimated occupancy will very likely differ (the stay assignment is probabilistic) since there are many more people in one census tract than the other. Furthermore, as previously stated the probability of assigning a person to a building depends also on the surrounding buildings.

Therefore, while the nominal building capacities are generated from standards, these are combined with empirical mobile phone data and the urban spatial structure to generate the estimated occupancy. We have clarified this sentence to emphasize that the occupancy we estimate is different from that based *solely* on building type.

2. Table 2 in the SI shows how points of interest are mapped to building types. There appears to be errors in this table. For example, spa, beauty salon, and haircare are all mapped to athletic facilities, while a stadium is mapped to a place of worship.

For each building type which doesn't obviously exist within the 16 different classes, we have made a best-rational-estimate as to which PCA category is most similar, however we realise that these choices are subjective and based largely on experience. As stated in Supplementary Note 2, for those building types with no reference in the dataset, PCAs were chosen based on maximum Occupancy Load Factors for fire evacuation as defined by the National Fire Protection Agency's NFPA 101-Fire Safety Code [A], adopted in the City of Boston's building codes.

It should be noted that where templates were unavailable from the DOE reference building sets, we excluded this building type from the urban energy model. However, we had to include all *open* tract buildings for the stay assignment process.

[A] National Fire Protection Association. Life safety code handbook: with the complete text of the 2015 edition of NFPA 101, Life safety code. 2015.

3. In section 4 of the SI, you recognize that expert energy modelers are used to model individual building energy use but explain that for the purpose of this paper a streamlined, automated approach was taken and a new tool developed using average building characteristics (commonly assigned through "archetypes"). Has this tool been validated against the results from

expert energy modelers? Other options of validation that you may want to consider are the EIA's Commercial Building Energy Consumption Survey and Residential Energy Consumption Survey. Another approach would be to look at the energy consumption data from utilities or energy authorities.

We thank the reviewer for pointing out the paper's lack of clarity in this regard and have updated the manuscript accordingly. Our baseline reference model results have compared to the national average EUIs for the equivalent building types in the Commercial Building Energy Consumption Survey and the Residential Building Energy Consumption Survey, and the average difference of the modeled EUI compared to the CBECS/RBECS averages ranged between 5% and 20% for the modeled building types in the geographic region of interest, with the simulated value always being higher than the CBECS reference [A].

Unfortunately, actual energy use data was unavailable at the resolution needed for a useful comparison.

[A] Carlos Cerezo Davila, Christoph F Reinhart, and Jamie L Bemis. Modeling boston: A workflow for the efficient generation and maintenance of urban building energy models from existing geospatial datasets. *Energy*, 117:237–250, 2016.

4. I'm a bit confused by which buildings are actually included in your analysis. When reading the main text, I thought all buildings were being classified and included. Then in the SI, it reads "we have chosen to only model the energy use of buildings falling within the classes available in the DOE dataset." What fraction of the actual buildings in the Back Bay neighborhood does this represent? How do these buildings differ from those that are not included in the study? Does this elimination of buildings in the analysis bias the results and, if so, by how much?

We agree with the reviewer that paper was unclear in this regard and indeed had some typos.

In the 5 tracts there are 1330 buildings which can potentially hold people. All of the open buildings are included for the people assignment, since the stay data is accurate at the tract level. Hence, leaving out buildings at this stage would likely lead to a systematic overestimation in building occupancy.

Out of these 1330 buildings, we were able to model 1,266 buildings using the DOE reference buildings. The other buildings were considered in the occupancy-assignment process (since they could still be open and attract people), however, due to atypical uses (fire/police, churches, garage etc), no appropriate reference model existed and therefore we opted not to include these buildings in the energy model. We note that for these building classes previous work has been unable to produce satisfactory results [A].

We have clarified in the manuscript that our conclusions only relate to the modeled building types. Since the majority of urban buildings are within the included types we do not feel that the results of the study are significantly diminished and we make no conclusion about the building types which we have not modelled.

[A] Carlos Cerezo Davila, Christoph F Reinhart, and Jamie L Bemis. Modeling boston: A workflow for the efficient generation and maintenance of urban building energy models from existing geospatial datasets. *Energy*, 117:237–250, 2016.

5. I'm a bit confused by your low and high impact calculations. These are defined by absolute and relative mobile occupancy. However, you only have data on 20% of the cell phone users in the area. I assume you are using TimeGeo to scale this data for the rest of the population and that these scaled values is what actually is going into Eqns 2-3. If this is the case, please clarify this in the text surrounding these equations. Additionally, if this is the case, your scaling will introduce uncertainty and this is not discussed. This should be quantified. The uncertainty in the model will influence the significance of the results.

The TimeGeo model uses the distribution of the observed mobility patterns to simulate the movements of the entire Boston metro population. Therefore, TimeGeo simulates 3.5 million users and the results align with best-in-class transportation demand models ([A] and Supplementary Note 1). The user visitation patterns are considered to be statistically identical to those of the real users at the census tract level [A].

Therefore, in the revised manuscript we have emphasized that TimeGeo agrees with best-in-class transportation models and is consistent with the population data in the census, which is untrue for urban models which use the DOE reference building models (see Fig. 2B in the revised manuscript). Therefore, our work represents an important first step towards realistic occupant presence modelling at an urban scale. Furthermore, while we agree that the two occupancy-scenarios could be improved, we consider them to be a rational approach for the two bounds of responsiveness to occupancy without any alternative available model that functions on this scale. It is hoped that our work will stimulate further development in the area of urban scale occupant behavior modeling.

[A] Jiang, S., Yang, Y., Gupta, S., Veneziano, D., Athavale, S., & González, M. C. (2016). The TimeGeo modeling framework for urban mobility without travel surveys. *Proceedings of the National Academy of Sciences*, 113(37), E5370-E5378.

6. Line 76: The mobile data is collected for a 6 week period in 2010. When were these 6 weeks? Were they consecutive weeks or did they include different seasons? Did they include holidays, such as Christmas, that would alter typical patterns of shopping? How significant was the variation found between weeks and between seasons?

The mobile phone data was collected over the period 20th February till March 30th 2010 and did not include major holidays. The Figure R1 below illustrates the regularity of the

data for the real users for the 5 complete weeks in the period, showing the fraction of all user-trips that were observed to take place within each particular hour for the active users, denoted as $P(t)$. Each trip represents a transition between stay points. The TimeGeo model then uses the observed mobility patterns of the active users during this period to simulate the movements of the entire Boston metro population for a typical weekday [A].

Figure R1: Fraction of all user trips within each hour over the observation period for active users.

We agree with the reviewer that the limited observation period is indeed a limitation of the model, which could be improved with longer data durations and higher resolution data. Accordingly, we are unable to quantify seasonal effects in our occupancy model. However, we stress that the population-wide mobility model based on the data is consistent with current best-in-class models for predicting typical transportation demands [A] and traffic conditions [B].

Furthermore, the purpose of this work is to understand how *typical* building occupancies estimated with empirical data compare to currently assumed values (which in urban energy models are based on the DOE reference buildings or from other estimates based on building types). Subsequently, given the huge discrepancy between the typical mobile-inferred occupancy and that of the reference buildings, our study certainly highlights that in general for most buildings, occupancy is likely to be significantly different to standard assumptions at most times. This has strong implications for predicting average energy consumption, which is highly relevant for the performance of ECMs.

[A] Jiang, S., Yang, Y., Gupta, S., Veneziano, D., Athavale, S., & González, M. C. (2016). The TimeGeo modeling framework for urban mobility without travel surveys. *Proceedings of the National Academy of Sciences*, 113(37), E5370-E5378.

[B] Çolak, S., Lima, A., & González, M. C. (2016). Understanding congested travel in urban areas. *Nature communications*, 7, 10793.

7. Lines 83-84: You assume the people only visit buildings within their own census tract. According to the Boston Globe (<https://www.bostonglobe.com/metro/2018/09/05/mass-average-commute-time-increases-longest-country/jt4VNcKMleeY1MGaxl3XYO/story.html>), Boston has the fifth longest commute to work in the country. Using the data that you have, you can test the validity of this assumption. In your set of active phone users, what fraction of users never left their census tract in the 6 week period? I would suggest reporting this fraction in the text to

validate this assumption, or, depending on the result, modifying this assumption. This is one of your assumptions that I think may radically alter your results. You mention in lines 144-146 that the mobile occupancy data differed by the DOE reference model by an order of magnitude at some times. I wonder if this discrepancy occurs since you only consider individuals staying in their residential census tract and miss all those who commute into the city for work.

We model the entire urban population and include population movements for all users with home locations in all the census tracts shown in Fig. 1A in the main text. Therefore, users will be mapped to a building within a tract *only at the times* they are in that tract for a stay. We have clarified this in the manuscript and have added Fig. 2 as shown below. Fig. 2A clearly shows why urban-scale mobility must be taken into account when considering the occupancy in a mixed-use urban district; some occupants who appear in the central mixed-use district have travelled very significant distances over the course of the day. Therefore, considering only more local occupants would introduce a significant error into the occupancy. We compare the implied number of people for this whole district in Fig. 2B – before occupancy for any individual buildings has been estimated using the mobile phone data – with the implied district occupancy using the reference buildings. We see that the model based on mobile data is much more reasonable in comparison with the census population than the reference buildings.

Figure 2: How many people travel-to and live-in to Back Bay? (A) Predicted journeys from the TimeGeo for users who visit the 5 tracts shown in Figure 1C. Each line represents approximately 50 users. (B) The total predicted hourly occupancy for the area shown in Figure 1C according to the TimeGeo model and as implied by the DORE reference building model. The census population for the area is also shown. (C) the predicted population flux as implied by the reference buildings and TimeGeo model.

8. I had a difficult time understanding Eqn. 1. What is the justification for raising the relative capacity to a factor of $3/2$? Has this attraction factor been validated? What is the uncertainty in this factor?

We agree with the reviewer that insufficient detail surrounding Equation 1 was given in the first version of the manuscript. Therefore, in the revised manuscript we provide much more clarity around Equation 1 and introduce the exponent as $(1+\mu)$, as well as including the sensitivity analysis to the parameter μ .

Our intuition for an exponent greater than 1 is to mimic the rich-get-richer effect in the popularity of places, *i.e.* we anticipate that areas with more capacity for occupants performing a particular type of activity will attract more people per unit floor area than those areas with less opportunity for that activity type. It is worth noting that this rich-get-richer effect has been observed in many different areas from city structure and growth to social networks [A]. The mobile phone data already has this effect at the census tract scale, since we observe that census tracts with large numbers of shops and restaurants attract more 'other type' stays. The parameter μ creates this effect on a city block scale within the census tracts and as a result, we get clustering of occupants where there are clusters of similar facilities. This leads to popular districts for certain types of activities – *i.e.* shopping or restaurant districts.

We included a study of the sensitivity of the resulting occupancy to the parameter μ . Expressing the exponent in Equation 1 as $(1+\mu)$ in the revised manuscript, we perform a sensitivity analysis in the range μ is 0-1. We believe this is a reasonable range since if $\mu < 0$ then districts with lots of opportunities will be less popular per unit floor space and $\mu > 1$ leads to unreasonably large occupancies in certain buildings. Details of this are provided in the main text and Supplementary Note 5. Although larger μ shifts the occupancy distribution for the buildings towards a few more highly occupied spaces, the effect of this parameter is small in comparison to the differences between the mobile-inferred and DOE reference occupancy.

[A] Perc, M. (2014). The Matthew effect in empirical data. *Journal of The Royal Society Interface*, 11(98), 20140378

Other comments:

9. Line 1: The use of the word “huge” is vague. Provide a more quantitative measure.

We have removed the word “huge” and added a more quantitative measure

10. Figure 1: I'm not sure what the black boxes in A and B represent. Is B the blow up of the black box in A, and C the blow up of the black box in B? The expansion factors, as indicated by color, in A cannot be discerned for the census tracts within the black box.

Yes the black boxes represent the domain of the next subfigure. We have clarified this in the manuscript. The outlines of the tracts have been removed for clarity in A.

Overall, I worry that assumption and model uncertainty and bias have not been well documented throughout the paper, making it difficult to interpret the validity of the results. I would strongly advise quantifying this uncertainty and bias prior to publication.

We thank the reviewer for their enlightening comments. We hope that the significant changes made in the revised manuscript as well as the extensive clarifications have eased the reviewers concerns.

Reviewers' comments:

Reviewer #1 (Remarks to the Author):

Dear authors, thanks for including the references to studies on statistical modeling for city-scale evaluations of energy efficiency measures, and studies on stochastic modeling of occupancy rates for building energy models, as such the current version of the paper is more specific and realistic in the definition of its potential contributions. Also, the analytical framework is better described than in the previous version of the paper.

1. What are the major claims of the paper?

In the discussion of the revised paper, the study correctly states that it improves urban scale building occupancy estimates. I agree that this is the main contribution of this study. However, in the abstract, the three major claims of this study are that it: (a) produced “a novel framework to estimate building occupancy at unprecedented scale,” (b) represents “an innovative application of mobile phone data,” and (c) is “a first step to connect urban building efficiency policies with building occupancy at a city scale”. These claims could be debated in the following way:

(a) the scale of the analysis has been tackled by other studies, such as Urban-EPC application in Manhattan (pages 447-469 in https://doi.org/10.1007/978-3-319-18368-8_24), and the proposed framework existed in a similar form in the software package called SunTool (Solar Energy 81 (2007) 1196–1211)

(b) mobile data has been used for occupancy estimates for building energy analyses (Energy and Buildings, Volume 47, April 2012, Pages 584-591, Building and Environment, Volume 141, 15 August 2018, Pages 1-15, and many more studies)

(c) this statement seems to be inaccurate based on the papers that are already cited

2. Are they novel and will they be of interest to others in the community and the wider field?

The novelty of this study is in the application that accounts for over a thousand buildings (1,266) in Boston, while applied/similar frameworks exist for smaller building subsets, with the exception of Urban-EPC’s application in Manhattan, which included energy simulations of 45,920 Manhattan buildings and a modification to typical occupancy profiles based on U.S. Census Bureau data.

3. If the conclusions are not original, it would be helpful if you could provide relevant references.

The conclusions assume that the proposed energy modeling results with mobile occupancy data improve the model accuracy without clear evidences that this might be the case. The study assumed that occupancy is the main driver for overall accuracy levels of building energy models. While the occupancy profiles are important, they are just one of several important factors for accuracy of building energy models (Applied Energy, Volume 182, 15 November 2016, Pages 115-134).

4. Is the work convincing, and if not, what further evidence would be required to strengthen the conclusions?

The paper hypothesizes that occupancy is the main or the only parameter required to improve accuracy of building energy models in cities. The paper either needs uncertainty analysis and error propagation to statistically define the quality of presented results for building energy simulations (not just sensitivity analyses of occupancy rates) or it needs validation with experimental data. In its current form, the paper presents a lot of work with limited ability to draw reliable conclusions. Importantly, the time-series of occupancy rates were averaged, rather than used in their most novel form as real continuous data. The authors claim that the averaged effect of time-series data is the most important one, but they are committing the same fallacy of producing averaged or typical occupancy profiles that they were trying to criticize and move forward from.

Thank you for presenting this work,

Jelena Srebric, Ph.D.

Professor, University of Maryland

Director, Cluster for Sustainability in the Built Environment (CITY@UMD)

Reviewer #2 (Remarks to the Author):

Thanks for addressing my and other reviewers' comments. The manuscript is much improved for clarity now. Although the reviewer values the methodology and work flow, I have reservation on the results especially the statement about phone data derived occupancy is likely to be 5 times lower than that used in DOE reference building models as their intended uses are different. Reference models are used for annual performance simulation with year-long 8760 hourly schedules of occupants, internal heat gains, HVAC operation etc. Reviewer hope future ground truth data can help validate the results.

Reviewer #3 (Remarks to the Author):

I appreciate the thoroughness of the authors' responses. The significant changes that were made to the revised manuscript has greatly clarified the research and underlying assumptions. I particularly appreciate the addition of Figure 2 and the sensitivity analysis of the non-linear parameter in Eqn 1.

I have a few minor suggestions for further clarification:

1. While you clarify the date range for the mobile phone data in your response, I cannot see this date range in the revised manuscript. In line 105, you mention that the data is collected for 6 weeks in 2010. I think your manuscript will be strengthened by including the date range so that the readers would not be left with the questions I had in comment #6.

2. Occasionally, terminology is used that, if you are trying to reach a broad scientific audience, I'm worried may not be familiar to the reader. The acronym "CDR" is used a few times in the manuscript but not defined. Another example, in line 112 you use "home stays." This could be interpreted as visits to your own home only or any residential home. Providing a short definition would be helpful.

3. In line 428, it appears that you have used "floor height" and "story height" interchangeably. If this is the case, I would suggest consistently using one term.

4. Be sure to cite Python packages, such as Eppy in line 469.

5. There were spelling errors (raecords in line 107), grammatical errors, and incorrect referencing of equations (line 177). The editorial staff should be well-equipped to fix these.

Overall I find the manuscript much improved and support the decision to publish this novel and interesting work.

Detailed Response to review comments

The reviewer comments are italicized and our responses are indented and highlighted in blue

We thank all the reviewers for their feedback which we believe has improved the quality of the manuscript. We have endeavored to provide a satisfactory response to all the reviewer comments.

Reviewer #1

Dear authors, thanks for including the references to studies on statistical modeling for city-scale evaluations of energy efficiency measures, and studies on stochastic modeling of occupancy rates for building energy models, as such the current version of the paper is more specific and realistic in the definition of its potential contributions. Also, the analytical framework is better described than in the previous version of the paper.

The authors thank the reviewer for their positive remarks and their suggestions, which have certainly benefited the work.

1. *What are the major claims of the paper?*

In the discussion of the revised paper, the study correctly states that it improves urban scale building occupancy estimates. I agree that this is the main contribution of this study. However, in the abstract, the three major claims of this study are that it: (a) produced “a novel framework to estimate building occupancy at unprecedented scale,” (b) represents “an innovative application of mobile phone data,” and (c) is “a first step to connect urban building efficiency policies with building occupancy at a city scale”. These claims could be debated in the following way:

(a) the scale of the analysis has been tackled by other studies, such as Urban-EPC application in Manhattan (pages 447-469 in https://doi.org/10.1007/978-3-319-18368-8_24), and the proposed framework existed in a similar form in the software package called SunTool (Solar Energy 81 (2007) 1196–1211)

(b) mobile data has been used for occupancy estimates for building energy analyses (Energy and Buildings, Volume 47, April 2012, Pages 584-591, Building and Environment, Volume 141, 15 August 2018, Pages 1-15, and many more studies)

(c) this statement seems to be inaccurate based on the papers that are already cited

We have made changes to the abstract to emphasize that our primary contribution is the improvement of occupancy estimates for urban scale building performance models. Regarding the novelty of the study, we believe that sophisticated building occupancy models have not previously been used in studies of this scale, and furthermore, existing

methods of improved occupancy modelling would be exceptionally difficult to apply at this scale.

In response to (a) we recognise that the scale of the analysis has been tackled before, including [1-3]. These studies have all used reference building occupancy profiles. We thank the reviewer for bringing the Urban-EPC study [3] to our attention, which uses reference building occupancy profiles but modifies occupant density based on population density data from the census. While this certainly represents an improvement, it cannot provide heterogeneous occupancy profiles for different buildings of the the same type and makes no effort to be aligned with transportation models. Furthermore, it will not conserve the number of occupants, whereas in our study, an occupant who leaves a building may become an occupant in another building. The SunTool study [4] is certainly an useful and influential tool for designing small neighborhoods. However, it uses the occupancy model of [5] which requires the initial number of occupants to be pre-specified for each building and therefore cannot be applied where the number of occupants using each building is unknown. Furthermore this occupancy model is calibrated using high resolution occupant presence data from a single office building (the LESO-PB building at the EPFL) [5].

In response to (b) we certainly agree with the reviewer that mobile phone based data has been used to estimate building occupancy, however we are unaware of any framework which is applicable to thousands of buildings and can be scaled to consider the visitations of millions of potential building occupants. The reviewer suggests two studies; the first cited study [6] considered two buildings on the MIT campus with wifi connectivity data and the second cited study [7] considered one building and a single monitoring point to the mobile network. Additionally, we note that using wifi data to infer occupancy (as in the first study) is likely to be very difficult to apply to the occupancy of thousands of diverse buildings. We thank the reviewer for drawing our attention to the latter study, however we argue that due to its use of a single building and single network monitoring cell this is fundamentally different to our proposed framework. One of the primary novelties of the TimeGeo framework [8], upon which our population mobility model is based, is the ability to model entire population movements based on empirical mobile phone data.

In response to (c), we have made several changes to the abstract to ensure that the claims of our paper are not overstated.

- [1] Chen Y, Hong T, Piette MA. Automatic generation and simulation of urban building energy models based on city datasets for city-scale building retrofit analysis. *Applied Energy*. 2017 Nov 1;205:323-35.
- [2] Davila, Carlos Cerezo, Christoph F. Reinhart, and Jamie L. Bemis. "Modeling Boston: A workflow for the efficient generation and maintenance of urban building energy models from existing geospatial datasets." *Energy* 117 (2016): 237-250.
- [3] Quan SJ, Li Q, Augenbroe G, Brown J, Yang PP. Urban data and building energy modeling: A GIS-based urban building energy modeling system using the urban-EPC engine. In *Planning Support Systems and Smart Cities 2015* (pp. 447-469). Springer, Cham.
- [4] Robinson D, Campbell N, Gaiser W, Kabel K, Le-Mouel A, Morel N, Page J, Stankovic S, Stone A. SUNtool—A new modelling paradigm for simulating and optimising urban sustainability. *Solar Energy*. 2007 Sep 1;81(9):1196-211.
- [5] Page J, Robinson D, Morel N, Scartezzini JL. A generalised stochastic model for the simulation of occupant presence. *Energy and buildings*. 2008 Jan 1;40(2):83-98.
- [6] Martani C, Lee D, Robinson P, Britter R, Ratti C. ENERNET: Studying the dynamic relationship between building occupancy and energy consumption. *Energy and Buildings*. 2012 Apr 1;47:584-91.
- [7] Pang Z, Xu P, O'Neill Z, Gu J, Qiu S, Lu X, Li X. Application of mobile positioning occupancy data for building energy simulation: An engineering case study. *Building and Environment*. 2018 Aug 15;141:1-5.
- [8] Jiang, S., Yang, Y., Gupta, S., Veneziano, D., Athavale, S., & González, M. C. (2016). The TimeGeo modeling framework for urban mobility without travel surveys. *Proceedings of the National Academy of Sciences*, 113(37), E5370-E5378.

2. *Are they novel and will they be of interest to others in the community and the wider field? The novelty of this study is in the application that accounts for over a thousand buildings (1,266) in Boston, while applied/similar frameworks exist for smaller building subsets, with the exception of Urban-EPC's application in Manhattan, which included energy simulations of 45,920 Manhattan buildings and a modification to typical occupancy profiles based on U.S. Census Bureau data.*

We again thank the reviewer for pointing out the Urban-EPC study [1], which is certainly relevant to our study. This Urban-EPC study uses reference building templates and proposes that occupancy is modified according to the census population density data. While this represents an improvement over the use of reference buildings alone, it cannot capture heterogenous occupancy profiles for different buildings of the same type in the same area, nor does it account for building occupants who move throughout the greater urban area over the course of a day. Given this, we believe that our study results

will be of interest to the building energy community and the wider field. In particular, we hope the work will stimulate greater interest in large-scale occupant behavior models which may tackle the uncertainty associated with the different occupant behavior scenarios highlighted in our work. We also hope that future large scale empirical datasets may become available, allowing rigorous validation of city-level occupancy models.

[1] Quan SJ, Li Q, Augenbroe G, Brown J, Yang PP. Urban data and building energy modeling: A GIS-based urban building energy modeling system using the urban-EPC engine. In *Planning Support Systems and Smart Cities 2015* (pp. 447-469). Springer, Cham.

3. If the conclusions are not original, it would be helpful if you could provide relevant references.

The conclusions assume that the proposed energy modeling results with mobile occupancy data improve the model accuracy without clear evidences that this might be the case. The study assumed that occupancy is the main driver for overall accuracy levels of building energy models. While the occupancy profiles are important, they are just one of several important factors for accuracy of building energy models (*Applied Energy*, Volume 182, 15 November 2016, Pages 115-134).

Our study highlights that the occupancy currently used in large scale urban energy simulations is likely to be inaccurate and is both inconsistent with census data and best-in-class transportation models. We then perform energy performance simulations to illustrate the effect of these inaccuracies in a state-of-the-art UBEM and demonstrate that, when occupancy which both agrees with transportation and population distribution models is used, this results in energy consumption predictions which are significantly lower than reference models in commercial buildings. Certainly, without experimental data it is impossible to know whether these exact energy predictions are more or less accurate, and as the reviewer correctly points out, in certain buildings occupancy is not a primary driver of energy use. However, occupancy is certainly regarded as one of the leading causes of error in the field of building energy modelling (several leading journals have dedicated entire special issues to understanding building occupancy, *i.e.* [1], and the International Energy Agency (IEA) has funded two Annexes dedicated to understanding occupant presence and behavior in buildings and improving models [2,3]), therefore by providing more accurate and heterogenous occupancy we feel our study makes a useful contribution to the field. Furthermore, the prediction range implied by the two impact-of-occupancy scenarios, which are rational approaches for the two

bounds of responsiveness to occupancy, illustrates that the impact of occupant behavior is likely to be highly significant and should be further studied.

[1] "Special issue on occupant behavior in buildings" Energy and Buildings Volume 133, 1 December 2016, Page 305. <https://doi.org/10.1016/j.enbuild.2016.09.027>

[2] International Energy Agency. IEA-EBC Annex 66: Definition and Simulation of Occupant Behavior in Buildings, 2018.

[3] International Energy Agency. IEA-EBC Annex 79: Occupant-Centric Building Design and Operation, 2018.

4. Is the work convincing, and if not, what further evidence would be required to strengthen the conclusions?

The paper hypothesizes that occupancy is the main or the only parameter required to improve accuracy of building energy models in cities. The paper either needs uncertainty analysis and error propagation to statistically define the quality of presented results for building energy simulations (not just sensitivity analyses of occupancy rates) or it needs validation with experimental data. In its current form, the paper presents a lot of work with limited ability to draw reliable conclusions. Importantly, the time-series of occupancy rates were averaged, rather than used in their most novel form as real continuous data. The authors claim that the averaged effect of time-series data is the most important one, but they are committing the same fallacy of producing averaged or typical occupancy profiles that they were trying to criticize and move forward from.

In order to strengthen the results regarding the energy consumption predictions, we have undertaken an extra set of simulations to illustrate the uncertainty in our occupancy assignments. These additional simulations consider the effects of the upper and lower bounds for the non-linear parameter μ , which affects how occupants are distributed among buildings in our model. Running energyplus simulations for all buildings with $\mu=0$, $\mu=0.5$ and $\mu=1$, we find that the effects of changing μ are much smaller than the differences between the high-impact and low-impact occupancy scenarios. We believe that this strengthens our conclusions, since the introduced uncertainty is much smaller than the range associated with the different occupant behavior scenarios.

We disagree that uncertainty analysis of parameters other than occupancy would be beneficial for this study, since the purposes of the paper are: (1) We highlight the differences in occupancy within urban energy models between standard methods based solely on reference building occupancy and our model based which includes empirical mobile phone data. (2) We demonstrate the effect of including this *heterogeneous* building-level occupancy in a large-scale Urban Building Energy Model. Therefore, we do not make any claim regarding the relative importance of including our proposed

heterogeneous building-level occupancy estimates compared with variations in other modeled building parameters, rather we simply demonstrate the range of implied energy use predictions and compare these with the standard method.

We also disagree with the assertion that we are committing the same fallacy of using average occupancy profiles, since our occupancy estimates are specific to individual buildings, representing a typical occupancy profile for each building. This contrasts to the current approach wherein occupancy profiles are estimated for *each building class* (i.e. each residence has the same occupancy profile). We have shown in the paper that these uniform occupancy estimates are inconsistent with both census and transportation data. Of course, if available, real continuous occupancy data would be preferable to our approach (which produces a unique typical occupancy profile for each building), however this data is not available covering a sufficient proportion of building occupants over a long enough timescale to be of use in the standard UBEM approach.

Reviewer #2

Thanks for addressing my and other reviewers' comments. The manuscript is much improved for clarity now. Although the reviewer values the methodology and work flow, I have reservation on the results especially the statement about phone data derived occupancy is likely to be 5 times lower than that used in DOE reference building models as their intended uses are different. Reference models are used for annual performance simulation with year-long 8760 hourly schedules of occupants, internal heat gains, HVAC operation etc. Reviewer hope future ground truth data can help validate the results.

We thank the reviewer for their positive comments and their input to the manuscript. We agree that the DOE reference buildings are an incredibly valuable resource for building energy performance simulation, and that they are often used in annual performance simulations and peak-load calculations, hence containing factors of safety. Urban Building Energy Models (UBEMs) which can be generated in semi-automated fashion are an emerging tool (i.e. [1-3]), with the potential to dramatically improve urban energy policy as it relates to urban buildings. In UBEMs, without empirically-based alternative occupancy data, these factors of safety in occupancy lead to an incorrect set of assumptions regarding city-scale building occupancy that neither agrees with transportation models or census data. Our goal in this work is to provide a framework which can generate realistic average occupancy profiles, which are heterogeneous on a building level, respecting census and transportation models. We find that in doing so our predicted occupancy levels are significantly lower than reference building occupancy

used in peak load calculation. We believe this represents an improvement for urban building energy modelling, where average occupancy rates are important to know for accurate predictions of annual energy use. We too hope that more ground truth data regarding large-scale building occupancy emerges which would be invaluable for validating these results and developing new and improved occupant presence and behavior models at urban-scale.

[1] Yixing Chen, Tianzhen Hong, and Mary Ann Piette. Automatic generation and simulation of urban building energy models based on city datasets for city-scale building retrofit analysis. *Applied Energy*, 205:323–335, 2017.

[2] P Nageler, G Zahrer, R Heimrath, T Mach, F Mauthner, I Leusbrock, H Schranzhofer, and C Hochenauer. Novel validated method for gis based automated dynamic urban building energy simulations. *Energy*, 139:142–154, 2017.

[3] Christoph Reinhart, Timur Dogan, J Alstan Jakubiec, Tarek Rakha, and Andrew Sang. Umi-an urban simulation environment for building energy use, daylighting and walkability. In 13th Conference of International Building Performance Simulation Association, Chambéry, France, 2013.

Reviewer #3

I appreciate the thoroughness of the authors' responses. The significant changes that were made to the revised manuscript has greatly clarified the research and underlying assumptions. I particularly appreciate the addition of Figure 2 and the sensitivity analysis of the non-linear parameter in Eqn 1.

We thank the reviewer for their positive remarks and agree that after a round of revisions the quality of the paper has significantly improved.

I have a few minor suggestions for further clarification:

1. While you clarify the date range for the mobile phone data in your response, I cannot see this date range in the revised manuscript. In line 105, you mention that the data is collected for 6 weeks in 2010. I think your manuscript will be strengthened by including the date range so that the readers would not be left with the questions I had in comment #6.

This date range has been clarified in the manuscript

2. Occasionally, terminology is used that, if you are trying to reach a broad scientific audience, I'm worried may not be familiar to the reader. The acronym "CDR" is used a few times in the manuscript but not defined. Another example, in line 112 you use "home stays." This could be interpreted as visits to your own home only or any residential home. Providing a short definition would be helpful.

We thank the reviewer for pointing out these oversights. Call Detail Records (CDR) and home stays have been defined at their first use. Home stays only represent visitations to your own home.

3. In line 428, it appears that you have used "floor height" and "story height" interchangeably. If this is the case, I would suggest consistently using one term.

Thanks for pointing this out. We have introduced floor height as the consistent nomenclature.

4. Be sure to cite Python packages, such as Eppy in line 469.

A citation has been added to the Eppy package.

5. There were spelling errors (raecords in line 107), grammatical errors, and incorrect referencing of equations (line 177). The editorial staff should be well-equipped to fix these.

Thanks for spotting these - we have fixed these errors.

Overall I find the manuscript much improved and support the decision to publish this novel and interesting work.

Once again, many thanks for your help with improving the manuscript.

REVIEWERS' COMMENTS:

Reviewer #1 (Remarks to the Author):

I appreciate that all my comments were taken seriously in this version of the paper, which was not the case in the previous revision.

Response to review comments

The reviewer comments are italicized and our responses are indented and highlighted in blue

Reviewer #1

I appreciate that all my comments were taken seriously in this version of the paper, which was not the case in the previous revision.

We would like to thank the reviewer for their constructive participation in the review process, and we feel that the paper has benefited greatly from their guidance. We hope that this work will stimulate further development in the field.